# Alveolar epithelial and vascular CXCR2 mediates transcytosis of CXCL1 in inflamed lungs

Katharina Thomas[1,9], Jan Rossaint[1,9], Nadine Ludwig[1], Sina Mersmann[1], Niklas Kötting[1], Julia Grenzheuser[1], Lena Schemmelmann[1], Marina Oguama [1], Andreas Margraf[1], Helena Block[1], Katharina Henke[1], Katharina Hellenthal [1], Valbona Mirakaj[2], Volker Gerke [3], Uwe Hansen[4], Karin Gäher[4], Miguel Engelhardt[5], Johannes Roth [6], Johannes Eble[5], Elin Hub[7], Antal Rot[7], Ronen Alon [8] & Alexander Zarbock [1] ✉

Pulmonary infections are characterized by neutrophil recruitment into the lung driven by chemokine ligands of CXCR2, which is expressed on neutrophils, but also present in non-hematopoietic lung cells, in which its role remains unclear. We hypothesize that CXCR2 in epithelial and endothelial cells contributes to neutrophil recruitment into the lung by modifying the availability of its cognate chemokines in lung alveoli. Using conditional endothelial and epithelial CXCR2 knockout mice, we demonstrate that selective CXCR2 deletion in either compartment impairs neutrophil recruitment into the lung during bacterial pneumonia and reduces bacterial clearance. We show that CXCR2 ablation in epithelial and endothelial cells compromises respective trans-epithelial and trans-endothelial transcytosis of alveolar CXCL1. Mechanistically, CXCR2-mediated CXCL1 endothelial and epithelial cell transcytosis requires the function of Bruton's tyrosine kinase in these cells. In conclusion, CXCR2 plays an important role in alveolar epithelial and endothelial cells, where it mediates cognate chemokine transcytosis, thus actively supporting their activities in neutrophil recruitment to the infected lungs.

Pulmonary inflammation occurs in a variety of clinical settings and is most commonly associated with bacterial and viral pneumonias, which remain major causes of morbidity and high mortality in both industrialized and developing countries[1], with latter, depending on the pathogen, ranging from 27% to 45%[2]. An effective host defense against bacterial pulmonary infections relies on the rapid clearance of the

pathogenic microorganism from the respiratory tract as initiated by innate immune responses. Polymorphonuclear neutrophils (PMN) play a key role in this process and constitute the first line of defense against invading pathogens[3]. Neutrophil migration into the lungs is a tightly controlled process of subsequent events, involving neutrophils passing the endothelial barrier of alveolar capillaries, a thin interstitial space, and

[1]Department of Anesthesiology, Intensive Care and Pain Medicine, University Hospital Münster, Münster, Germany. [2]Department of Anaesthesiology and Intensive Care Medicine, University Tübingen, Tübingen, Germany. [3]Institute of Medical Biochemistry, Centre for Molecular Biology of Inflammation (ZMBE), University of Münster, Münster, Germany. [4]Institute of Experimental Musculoskeletal Medicine, University Hospital Münster, Münster, Germany. [5]Institute of Physiological Chemistry and Pathobiochemistry, University of Münster, Münster, Germany. [6]Institute of Immunology, University Hospital Münster, Münster, Germany. [7]Centre for Microvascular Research, William Harvey Research Institute, Barts and The London School of Medicine and Dentistry, Queen Mary University of London, London, UK. [8]Department of Immunology and Regenerative Biology, Weizmann Institute of Science, Rehovot, Israel. [9]These authors contributed equally: Katharina Thomas, Jan Rossaint. ✉e-mail: zarbock@uni-muenster.de

afterwards the alveolar epithelial layer to reach the alveolar space[4]. Uncontrolled neutrophil activation and recruitment result in increased endothelial and epithelial permeability, leading to alveolar flooding with a protein-rich edema fluid. In particular, increased endothelial permeability results from endothelial cell activation, a state of heightened responsiveness to leukocytes that is induced by various stimuli. Endothelial cell activation is categorized as either type I, more rapid entailing responses that are independent of gene expression or type II, a slower one that depend on new gene expression (type II activation)[5]. In inflammation chemokines are produced by practically every cell, including neutrophils themselves and play a crucial role in driving and directing the migration of immune cells to sites of infection or inflammation[6].

Chemokine transcytosis during acute inflammation involves the transport of chemokines across endothelial barriers to facilitate immune cell recruitment in response to inflammatory stimuli. Chemokine binding to classical 7-transmembrane receptors (7TMR) may trigger G-protein-coupled activation of intracellular signaling cascades. In contrast, non-signaling 7TMRs do not activate G-protein-coupled signaling but may foster chemokine binding and receptor-mediated endocytosis, involving activation of MAP kinases and ß-arrestins[7]. Once internalized, chemokines undergo intracellular trafficking and vesicular transport across the endothelial barrier, a process known as transcytosis. Upon reaching the tissue space, they establish gradients that guide leukocytes to the site of inflammation, amplifying the immune response.

ACKR1 (Atypical Chemokine Receptor 1) has been shown to decisively contribute to the endothelial cell binding, internalization, and transcytosis of chemokines, thus supporting their function in leukocyte extravasation[8,9]. However, within the vascular tree ACKR1 is normally expressed only in the postcapillary and collective venules and small veins and is conspicuously missing in the capillaries throughout all organs and tissues[10], including alveolar capillaries[11,12]. Therefore, ACKR1 cannot contribute to the chemokine transcytosis and immobilization in the alveolar vascular network and thus support chemokine driven leukocyte emigration into the alveolar spaces. Thus, we set to investigate if an alternative chemokine transcytosis pathway exists in the alveolar vessels.

CXCR2 is the main functional chemokine receptor expressed on neutrophils but also by lung endothelial and epithelial cells[13–16]. Its main ligands are chemokines, CXCL1, CXCL2, CXCL3, CXCL5, CXCL6, CXCL7, and in human CXCL8, which are involved in the recruitment and activation of neutrophils at inflammatory sites[17,18]. All known CXCR ligands signal through the receptor's G protein subunits ($G\alpha_{i1}$, $G\alpha_{i2}$, $G\alpha_{i3}$, $G\alpha_{oA}$, $G\alpha_{oB}$, and $G\alpha_{15}$) without a relevant individual, ligand-specific binding towards any one type of G protein subunits[19]. Blocking or ablating CXCR2 leads to reduced neutrophil recruitment and diminished tissue damage in several animal models of inflammations[20,21]. Hematopoietic CXCR2 is key for innate immunity in infection, as its depletion can lead to massive bacterial overgrowth with fatal for the host consequences[22]. The study of LPS-induced changes in vascular permeability and neutrophil functional behavior suggested that CXCR2 expressed in the non-hematopoietic lung cells also contributes to the chemokine-driven inflammatory pathology[23]. However, the mechanisms by which endothelial and epithelial CXCR2 might facilitate neutrophil migration into the lung remain unclear.

In this study, we use a spectrum of in vivo and in vitro assays to show that non-hematopoietic CXCR2 expressed by epithelial and endothelial cells in the lung promotes efficient recruitment of neutrophils from the vascular compartment by mediating the transcytosis of its cognate ligands. This is the first description of the atypical function of CXCR2 in transcytosis and the first direct demonstration of such activity by any classical chemokine GPCR.

## Results
### CXCR2 on haematopoietic and non-haematopoietic cells is required for optimal host defense during bacterial pneumonia
To investigate the role of haematopoietic and non-haematopoietic CXCR2 in the lung, neutrophil recruitment and bacterial clearance were analyzed in global (CXCR2$^{-/-}$) as well as endothelial (CXCR2$^{fl/fl}$Cdh5$^{Cre+}$) and epithelial (CXCR2$^{fl/fl}$Shh$^{Cre+}$) cell type-specific conditional KO mice in a murine model of *K. pneumoniae*-induced pneumonia. The absence/expression of CXCR2 from all lung cells in CXCR2$^{-/-}$ (global knockout) mice as well as cell type-specific absence in lung endothelial cells in CXCR2$^{fl/fl}$Cdh5$^{Cre+}$ and epithelial cells in CXCR2$^{fl/fl}$Shh$^{Cre+}$ mice was validated by flow cytometry following IL-1β stimulation and found to be increased in WT mice after pneumonia induction (Supplementary Fig. 1). Isotype controls in conditional mouse strains were always performed in Cre-negative mice. Conditional CXCR2 knockout in endothelial and epithelial cells did not affect CXCR2 expression on circulating leukocytes (Supplementary Fig. 2). The analysis of baseline leukocyte counts in CXCR2$^{-/-}$, CXCR2$^{fl/fl}$Shh$^{Cre+}$ and CXCR2$^{fl/fl}$Cdh5$^{Cre+}$ showed no significant differences (Supplementary Fig. 3A-C). In addition, the surface expression of various cell surface markers and the intracellular expression of molecules involved in endocytosis and transcytosis pathways did not differ on pulmonary endothelial and epithelial cells isolated from the experimental mouse strains (Supplementary Fig. 4A-C). Interestingly, by using immunofluorescence staining, we did not observe expression of the atypical chemokine receptor ACKR1 in lung alveoli, but only in a subset of venules of the bronchial circulatory tree and larger veins with ACKR1 immunoreactivity detected on endothelial cells of alveolar capillaries or other segments of the pulmonary circulatory tree (Supplementary Fig. 4D-E).

Intratracheal instillation of *K. pneumoniae* resulted in recruitment of neutrophils into the lungs of WT control mice compared to saline-treated control mice and was significantly reduced in all CXCR2 knockout strains tested and WT that received a blocking CXCR2 antibody intratracheally (Fig. 1A, D, G and Supplementary Fig. 5A–I). The baseline and post-infection experiments were conducted concurrently. Impaired neutrophil recruitment in knockout mice and after intratracheal instillation of a blocking CXCR2 antibody was observed in the whole lung tissue and also in the bronchoalveolar lavage (BAL, Fig. 2A, D, G and Supplementary Fig. 5A-I). In all knockout mouse strains and WT mice who received a blocking CXCR2 antibody intratracheally, a decrease in neutrophil recruitment was associated with a significant increase in bacterial burden, as e.g., determined by the quantification of bacterial colony forming unites (CFU), in the lung and the spleen (Fig. 1B, C, E, F, H, I and Supplementary Fig. 5A-I). This effect was further evident in the BAL and the blood (Fig. 2B, C, E, F, H, I). The concurrent intratracheal administration of a non-blocking control antibody did not show any significant effects on neutrophil recruitment and CFU counts (Supplementary Fig. 5A-I). As a proof-of-concept, we additionally induced bacterial pneumonia by intratracheal instillation of *E. coli* (ATCC strain 25922) instead of K. pneumoniae and observed similar results (Supplementary Fig. 6). Administration of a higher CFU of K. pneumonia in the intratracheal inoculum led to a significantly decreased survival of CXCR2$^{-/-}$ as well as conditional CXCR2$^{fl/fl}$Cdh$^{Cre+}$ and CXCR2$^{fl/fl}$Shh$^{Cre+}$ mice specifically lacking CXCR2 in endothelial or epithelial cells compared to the respective control mice (Fig. 3A-C).

Total protein contents in the BAL, as a surrogate parameter for vascular permeability, were significantly elevated in global CXCR2$^{-/-}$ mice, but there were no differences in endothelial and epithelial specific CXCR2 knockout mice as compared to control WT mice (Supplementary Fig. 7A-C). Likewise, the intratracheal co-administration of CXCL1 (MW 8 kD) together with FITC-labeled Dextran (MW 10 kD) showed no significant extravasation of FITC-Dextran into the circulation in vivo, thus excluding the possibility of CXCR2 transport by increased vascular permeability rather than transcytosis (Supplementary Fig. 7D-E). To investigate a possible effect of CXCR2 knockout on CC chemokine homeostasis, we investigated trafficking of inflammatory monocytes in the lung in saline-treated control animals and after induction of pneumonia and did not observe significant differences (Supplementary Fig. 8A-F).

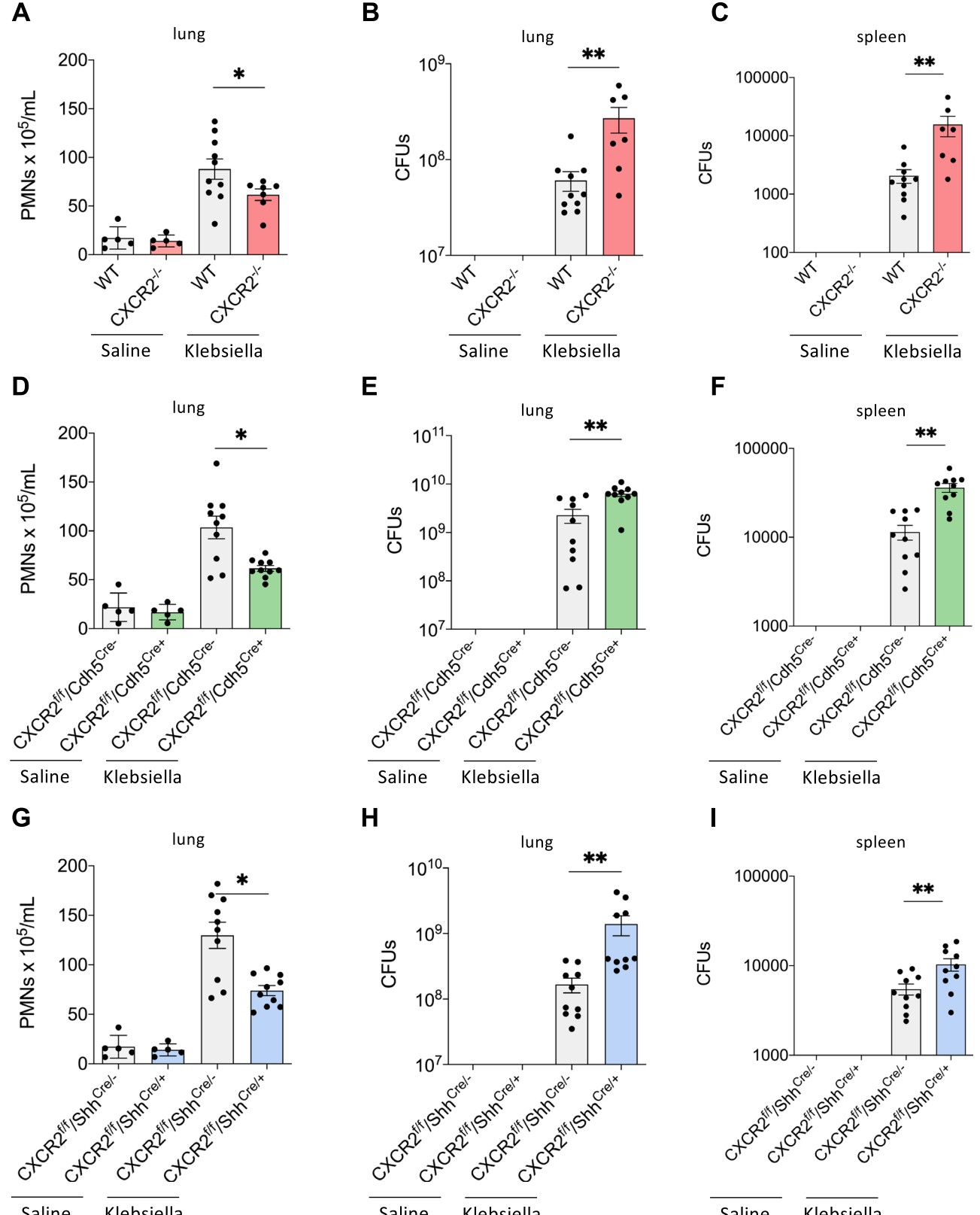

**Fig. 1 | Haematopoietic and non-haematopoietic CXCR2 is required for host defense during bacterial pneumonia.** CXCR2 KO mice (red = CXCR2⁻/⁻, green = CXCR2^fl/fl^Cdh5^Cre+^, blue = CXCR2^fl/fl^Shh^Cre+^) and appropriate WT/Cre⁻ control (gray) mice were intratracheally administered with sterile saline or with viable *K. pneumoniae*. Neutrophil recruitment into the lung (**A**, **D**, **G**), as well as CFUs in the lung (**B**, **E**, **H**) and spleen (**C**, **F**, **I**) were assessed 24 h after infection. Data are mean ± SD, $n$ = 5–10 biologically independent mice pre group, age 8–16 weeks, equal gender distribution, one-way ANOVA followed by Bonferroni correction, *$p < 0.05$, **$p < 0.01$.

Furthermore, we analyzed CXCL1 chemokine levels in plasma and BALF obtained from *K. pneumonia*-treated mice and found significantly increased chemokine levels upon pneumonia compared to baseline conditions (Supplementary Fig. 8G-L). Interestingly, CXCL1 levels were increased in plasma and BALF from CXCR2$^{-/-}$, CXCR2$^{fl/fl}$Cdh5$^{Cre+}$, and CXCR2$^{fl/fl}$Shh$^{Cre+}$ mice compared to control mice (Supplementary Fig. 8G-L). Endothelial CXCR2 deficiency led to CXCR2 desensitization by decreased CXCR2 expression on neutrophils (Supplementary

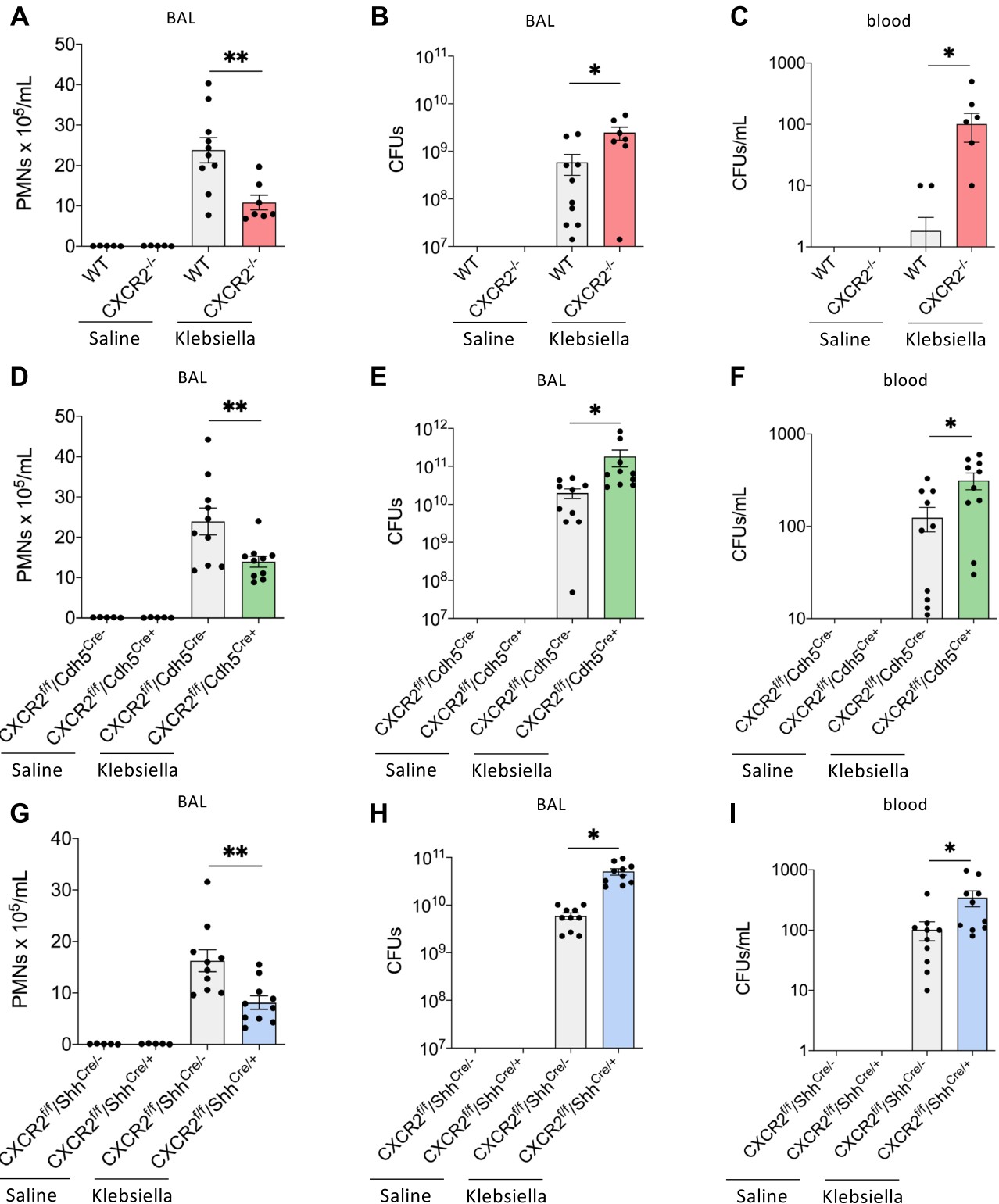

**Fig. 2 | Endothelial and epithelial CXCR2 are required for host defense during *K. pneumoniae*-induced pneumonia.** CXCR2 KO mice (red=CXCR2$^{-/-}$, green=CXCR2$^{fl/fl}$Cdh5$^{Cre+}$, blue=CXCR2$^{fl/fl}$Shh$^{Cre+}$) and appropriate WT/Cre$^-$ control (gray) mice were intratracheally administered with sterile saline or with viable *K. pneumoniae*. Neutrophil recruitment into BAL (**A**, **D**, **G**), as well as CFUs in the BAL (**B**, **E**, **H**) and blood (**C**, **F**, **I**) were assessed 24 h after infection. Data are mean ± SD, $n = 5$–10 biologically independent mice pre group, age 8–16 weeks, equal gender distribution, one-way ANOVA followed by Bonferroni correction, *$p < 0.05$, **$p < 0.01$.

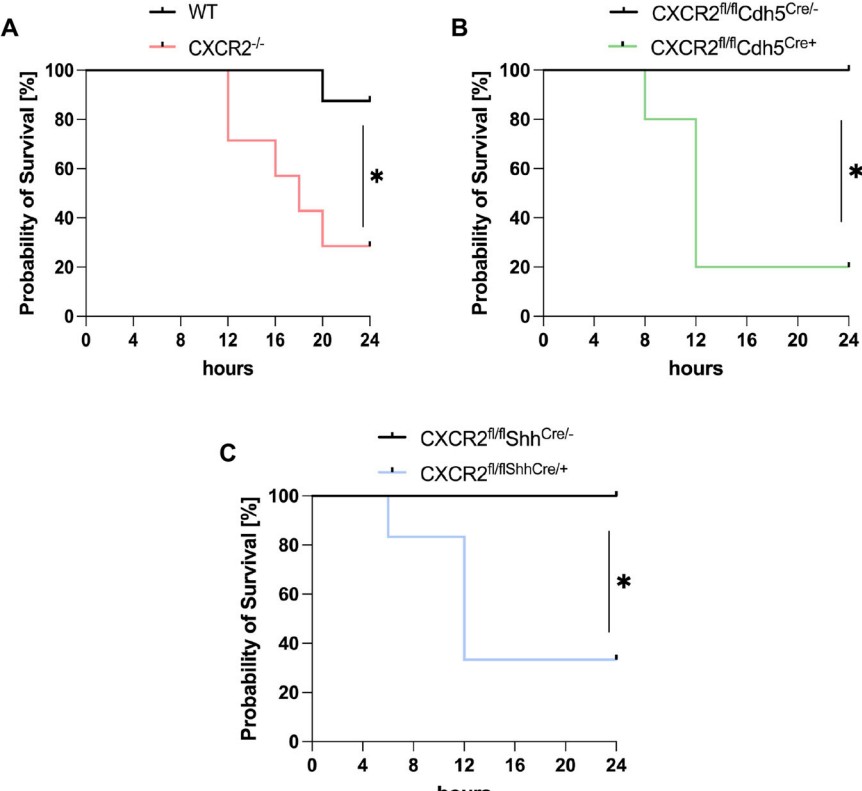

**Fig. 3 | Loss of endothelial and epithelial CXCR2 leads to reduced survival during *K. pneumoniae*-induced pneumonia.** CXCR2 KO mice (red=CXCR2⁻/⁻, green=CXCR2^fl/fl^Cdh5^Cre+^, blue=CXCR2^fl/fl^Shh^Cre+^) and appropriate WT/Cre⁻ control (gray) mice were intratracheally administered with a lethal CFU dose of viable *K. pneumoniae*. **A–C** Survival was analyzed 24 h. Data are mean ± SD, $n = 5$–8 biologically independent mice pre group, age 8–16 weeks, equal gender distribution, Kaplan-Meier analysis, *$p < 0.05$.

Fig. 8M) and subsequently impaired neutrophil migration 24 h after the induction of pneumonia (Supplementary Fig. 8N-P).

## CXCR2 on pulmonary endothelial and epithelial cells supports CXCL1-dependent neutrophil recruitment

The role of pulmonary endothelial and epithelial CXCR2 in the recruitment of neutrophils upon chemoattractant sensing was analyzed by intratracheal injection of CXCL1 or fMLP and visualization of neutrophils in the alveoli via intravital microscopy of the lung after 4 h (Fig. 4A, F, K). Whereas CXCL1 is a CXCR2 ligand, the chemotactic peptide fMLP does not bind to CXCR2 but acts via binding to the receptors FPR1 and FPR2. The counts of transmigrated neutrophils in the alveolar space were determined by flow cytometry. Intravital microscopy revealed a decreased number of transmigrating neutrophils in all knockout mice strains after CXCL1 stimulation (Fig. 4B, G, L) which also led to reduced neutrophil counts in the BAL (Fig. 4D, I, N). However, neutrophil recruitment was not altered in knockout mice after injection of the bacterial chemoattractant fMLP (Fig. 4C, E, H, J, M, O). Both CXCL1 and fMLP instillation did not significantly alter the thickness of the alveolar septa in the lung compared to saline-treated control mice after 4 h (Supplementary fig. 9A). To exclude that the used Gr1 antibody (clone RB6-8C5) leads to co-labeling of monocytes, we performed additional lung intravital microscopy experiments with an Alexa568-labeled anti-CD115 antibody specifically labeling monocytes after CXCL1 instillation and could not detect significant numbers of monocytes in the lung after 4 h (Supplementary fig. 9B). To exclude that neutrophil depletion using the anti-Gr1 antibody clone RB6-8C5 causes considerable co-depletion of monocytes, we injected CXCL1 intratracheally, administered the Gr1 antibody or isotype control and analyzed the abundance of neutrophils and monocytes in the lung after 4 h (Supplementary Fig. 9C-D). Furthermore, the administration of labeled Gr1

antibody did not significantly decrease neutrophil and monocyte blood counts in WT mice after the early observation period of 4 h (Supplementary Fig. 9E-F).

To visualize the precise allocation of neutrophils in different compartments of the lung, we performed spinning disc confocal microscopy of viable precision cut lung slices ex vivo (Supplementary Fig. 9G-H). These data demonstrate that CXCR2-deficiency in lung epithelial or endothelial cells leads to a decreased transmigration of neutrophils from the vascular compartment into the lung tissue (Supplementary Fig. 9I).

To assess if the impact of endothelial CXCR2 on neutrophil emigration is lung specific, neutrophil recruitment was analyzed in the murine cremaster muscle of endothelial specific CXCR2 knockout mice (CXCR2^fl/fl^Cdh5^Cre+^) and appropriate WT controls by intravital microscopy. For this purpose, mice were challenged with either an intrascrotal injection of TNF or with a local superfusion of CXCL1 (for 2 h) or fMLP (for 5 min) before analyzing neutrophil adherence and emigration. The ablation of endothelial CXCR2 did not affect neutrophil adhesion or emigration in the cremaster muscle following TNF-stimulation, fMLP or CXCL1 superfusion (Fig. 5A-F). Furthermore, the effect of endothelial CXCR2 deficiency in an in vivo mouse model of renal ischemia reperfusion injury (IRI) was determined. In consistence with previous results, neutrophil infiltration into the kidney as well as kidney function was not significantly altered in endothelial specific knockout mice (Fig. 5G, H). These results support the organ-specific function of pulmonary non-haematopoietic CXCR2 in supporting neutrophil trafficking into alveoli.

## Ablation of endothelial and epithelial CXCR2 decreases in vitro transmigration of neutrophils through an endothelial-epithelial bilayer cell construct

The influence of non-hematopoietic CXCR2 on neutrophil transmigration was analyzed in vitro by establishing a Transwell™

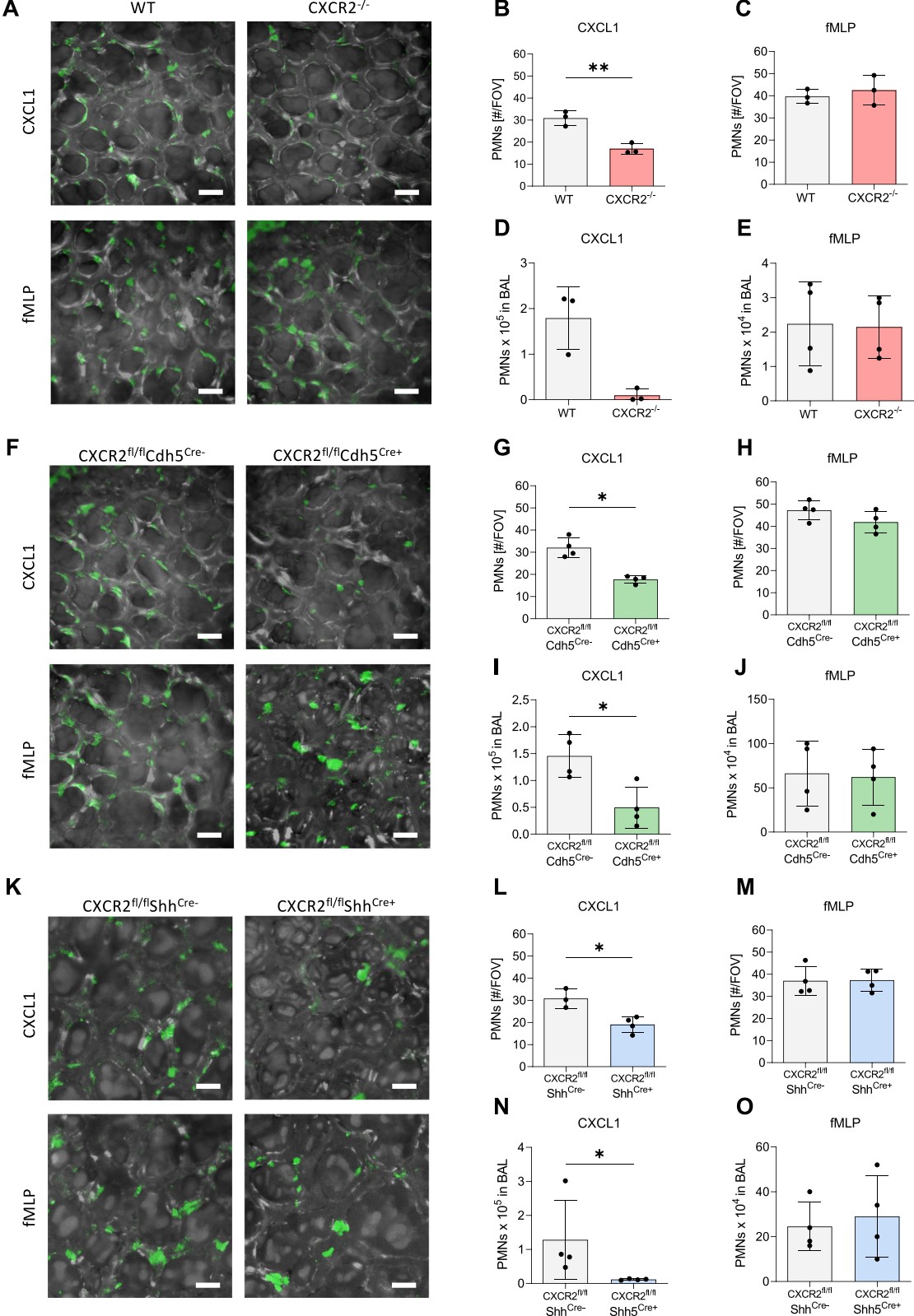

**Fig. 4 | Neutrophil recruitment into the alveoli is diminished in CXCR2 deficient mice upon CXCL1 but not fMLP treatment.** Intravital microscopy of the lung was performed in CXCR2 KO mice (red = CXCR2$^{-/-}$, green = CXCR2$^{fl/fl}$Cdh5$^{Cre+}$, blue = CXCR2$^{fl/fl}$Shh$^{Cre+}$) and appropriate WT/Cre$^-$ controls (gray) 4 h after intratracheal injection of CXCL1 or fMLP. Neutrophils were visualized in the alveoli by intravenous administration of an AF488-coupled Gr-1 antibody before starting microscopy (**A**, **F**, **K**). Neutrophil numbers were determined per field of view after CXCL1 (**B**, **G**, **L**) or fMLP (**C**, **H**, **M**) treatment. BALF was collected, and neutrophil infiltration into the alveoli was assessed by flow cytometry after CXCL1 (**D**, **I**, **N**) or fMLP (**E**, **J**, **O**) stimulation. Data are mean ± SD, scale bars equal 20 μm, $n$ = 3–4 biologically independent mice pre group, age 8–16 weeks, equal gender distribution, t-test, *$p < 0.05$, **$p < 0.01$.

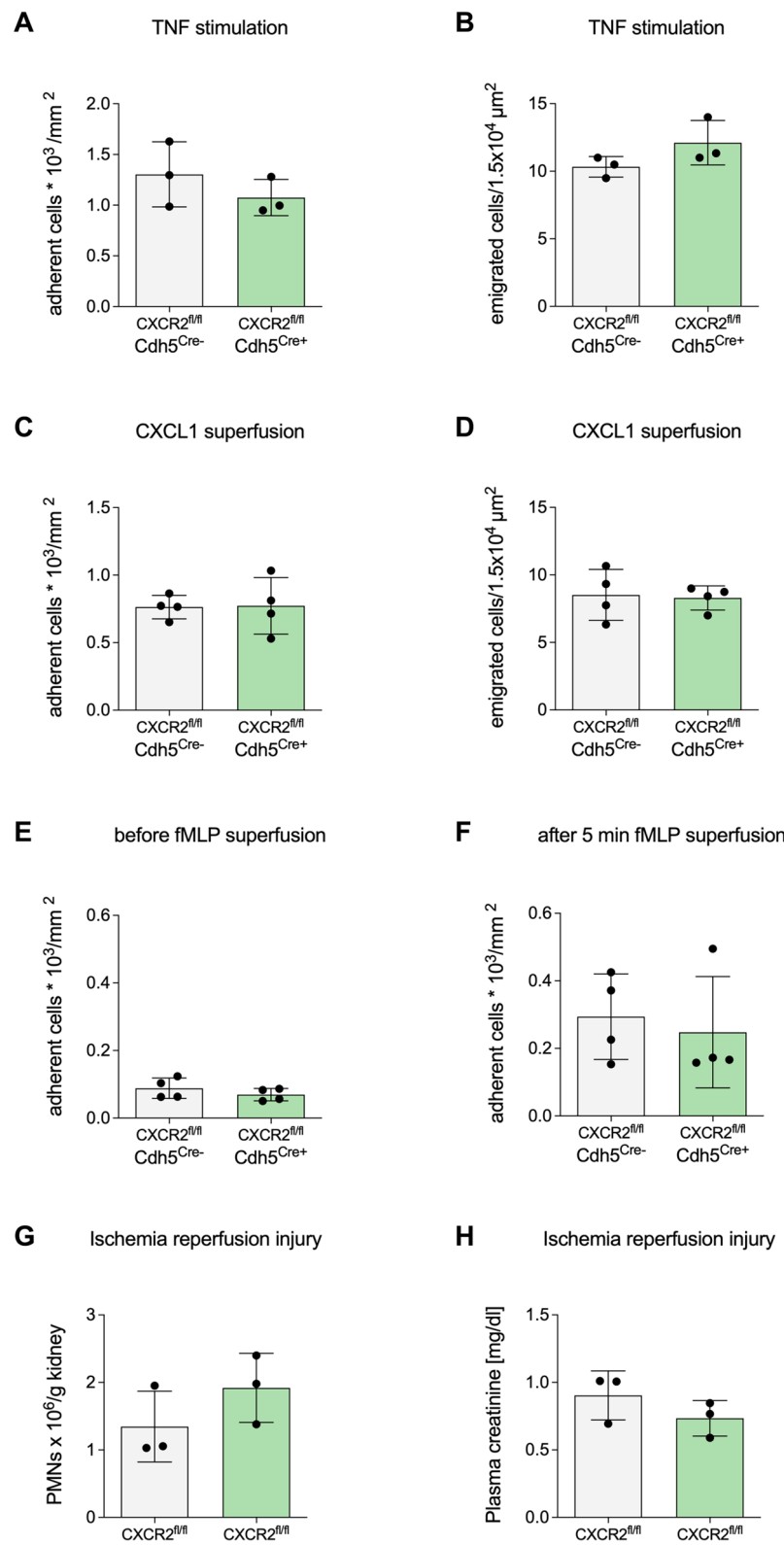

**Fig. 5 | Neutrophil recruitment in cremaster and kidney tissue is not altered in CXCR2fl/flCdh5Cre+ mice.** CXCR2fl/flCdh5Cre+ and Cre− mice were analyzed for neutrophil recruitment after acute inflammation of the cremaster muscle (**A**–**F**). Adherent and emigrated cells were analyzed after TNF stimulation (**A**, **B**) and CXCL1 superfusion of the cremaster muscle (**C**, **D**) or before and after fMLP superfusion (**E**, **F**). Neutrophil recruitment into the kidney (**G**) and plasma creatinine (**H**) were evaluated in CXCR2fl/flCdh5Cre+ and Cre− control mice after ischemia reperfusion injury (IRI). Data are mean ± SD, $n$ = 3–4 biologically independent mice pre group, age 8–16 weeks, only male mice for cremaster IMV, all other equal gender distribution, t-test.

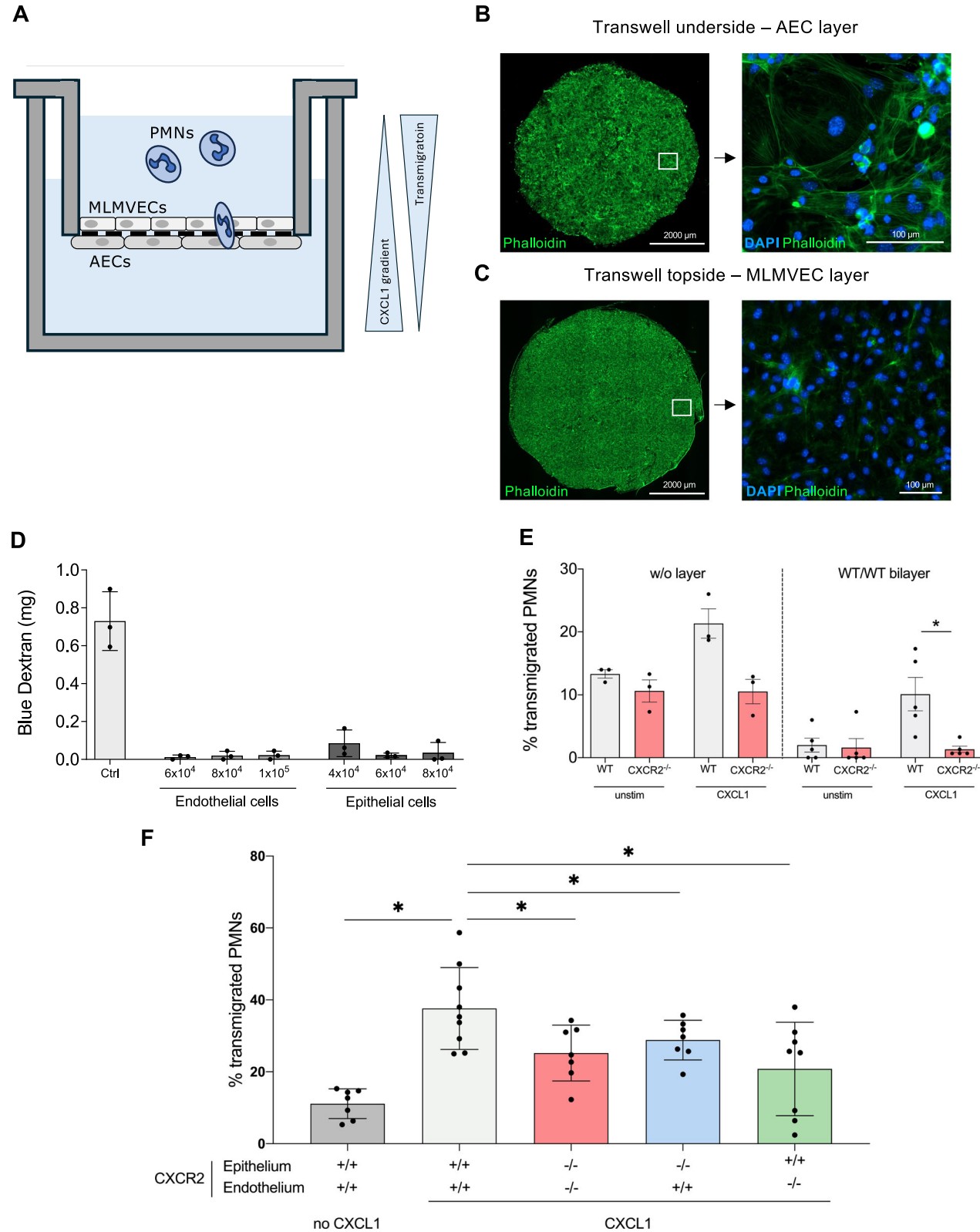

endothelial-epithelial bilayer model reflecting on the anatomical situs in murine lung alveoli (Fig. 6A). Confluency of primary murine lung microvascular endothelial (MLMVEC) and alveolar epithelial cells (AEC) on the transwells was ensured in fluorescent microscopy and measuring dextran diffusion across these monolayers. Both assays demonstrated intact cell layers suitable for the assay procedure (Fig. 6B-D). To prove the assay's reliability, the transmigration rate of WT and CXCR2$^{-/-}$ neutrophils through a bilayer

consisting of WT endothelial and WT epithelial cells towards a CXCL1 gradient was determined. WT neutrophils migrated through the layer upon CXCL1, whereas CXCR2$^{-/-}$ neutrophils failed to migrate and remained in the upper Transwell™ compartment (Fig. 6E). The impact of non-hematopoietic CXCR2 was finally analyzed by seeding endothelial and epithelial cells from WT and CXCR2$^{-/-}$ mice in different combinations onto the Transwell™ inserts and analyzing WT neutrophil transmigration towards a

**Fig. 6 | Loss of endothelial and epithelial CXCR2 decreases transmigration of neutrophils through an endothelial-epithelial bilayer in vitro.** Primary murine lung microvascular endothelial cells (MLMVEC) and alveolar epithelial cells (AEC) were cultured on different sides of a Transwell™ insert. Transmigration of WT bone marrow-derived neutrophils towards a CXCL1 gradient was analyzed in case of different WT and CXCR2$^{-/-}$ layer combinations (**A**). Confluency was assured via Phalloidin staining of (**B**) AECs and (**C**) MLMVECs on Transwell™ membranes. **D** To further confirm the development of confluent and tight cell layers a Blue Dextran solution was transferred into the upper Transwell™ chamber and allowed to distribute for 3 h at 37 °C. The amount of Blue Dextran in the lower chamber was

assessed in a plate reader. **E** Reliable transmigration of neutrophils was evaluated by determining transmigration efficiency of WT and CXCR2$^{-/-}$ neutrophils towards a CXCL1 gradient through either the Transwell membrane (w/o layer) or a bilayer consisting of WT AECs and MLMVECs (WT/WT bilayer). Transwells were equipped with WT/WT control layer or with CXCR2 deficient epithelial and endothelial layers, epithelial WT and endothelial KO layers or epithelial KO but endothelium WT layers (**F**). The percent of transmigrated neutrophils was assessed by the help of a Sysmex haemocytometer. Data are mean ± SD, $n = 3–7$ biologically independent samples, one-way ANOVA followed by Bonferroni correction, *$p < 0.05$.

---

CXCL1 gradient. A bilayer consisting of endothelial and epithelial cells both derived from CXCR2$^{-/-}$ mice resulted in a significantly reduced neutrophil transmigration compared to a WT/WT bilayer (Fig. 6F). A similar effect was observable in case of a WT endothelium but CXCR2$^{-/-}$ epithelium and vice versa (Fig. 6F). Of note, two populations appeared to be present in the group with WT epithelium and CXCR2$^{-/-}$ endothelium. However, this discrepancy was neither statistically significant within the group nor when tested against the other groups.

### Immune electron microscopy of CXCR2-mediated CXCL1 transcytosis through epithelial and endothelial cells

The dissemination of CXCR2 and CXCL1 in murine pulmonary endothelial and epithelial cells following infection with *K. pneumoniae* was analyzed by transmission electron microscopy. In ultrathin lung sections of *K. pneumoniae* infected WT mice CXCL1 clustered with both CXCR2 and clathrin in both epithelial and endothelial cells of the lung (Fig. 7A, D). CXCL1/CXCR2/clathrin complexes were observed apically and basally in epithelial and endothelial cells (Supplementary Fig. 10A-B). Furthermore, CXCL1 and CXCR2 appeared on tips of endothelial protrusions extending into the luminal space (Supplementary Fig. 10C). As expected, CXCR2 was missing in global knockout mice, whereas endogenous CXCL1 was still apparent in all cell layers, partly co-localized with clathrin (Fig. 7A). In case of an epithelial CXCR2 knockout, CXCR2 is indeed only present in endothelial cells and an endothelial specific knockout resulted in CXCR2 expression only in epithelial but not endothelial cells (Fig. 7B-C). Lung sections of Cre controls mice demonstrated an even distribution of CXCR2, CXCL1, and clathrin in endothelial and epithelial layers, which were comparable with WT conditions. Quantitative analysis of the images revealed that CXCL1 levels in KO mice were similar across all KO mouse strains and comparable to WT mice at baseline (Ctrl), thus likely representing cell-endogenous CXCL1 levels (Fig. 7E-F). To further elaborate on the exact organellar route or transcytosis, we performed transmigration assays using the transwell bilayer model with inhibitors of clathrin- (pitstop 2), caveolin- (MBCD), or dynamin-mediated transcytosis (dyngo 4a) and observed a significant reduction in neutrophil transmigration after inhibition of clathrin-mediated transcytosis (Fig. 8A) as well as reduced CXCL1 levels in the upper well above the endothelial cell layer (Fig. 8B).

### Reduced in vivo transcytosis of CXCL1 in the absence of endothelial and epithelial CXCR2

Next, we injected streptavidin-coupled CXCL1 intratracheally into the lungs of global (CXCR2$^{-/-}$) as well as endothelial (CXCR2$^{fl/fl}$Cdh5$^{Cre+}$) and epithelial (CXCR2$^{fl/fl}$Shh$^{Cre+}$) specific conditional KO mice with respective WT controls. After 4 h, blood samples were obtained and the levels of transcytosed streptavidin-coupled CXCL1 were analyzed by an ELISA using a CXCL1 capture antibody and a biotin-HRP conjugated detection antibody. Global (CXCR2$^{-/-}$) mice, endothelial (CXCR2$^{fl/fl}$Cdh5$^{Cre+}$) and epithelial (CXCR2$^{fl/fl}$Shh$^{Cre+}$) specific conditional KO mice showed a decreased transcytosis of CXCL1 in vivo (Fig. 9A). Of note, the expression of the endocytosis/transcytosis-related molecules clathrin, caveolin and dynamin was not significantly different among the different knockout mouse strains (Fig. 9B). It has

been previously demonstrated that the tyrosine kinase Btk (Bruton's tyrosine kinase) is involved in various intracellular signaling and transport pathways that mediate the cell's response to stimulation with various conserved inflammatory pathogen motifs and shapes the response to chemokine stimulation[24]. To analyze if CXCL1 transcytosis requires the presence of Btk in non-hematopoietic cells of the lung tissue, WT and Btk-KO recipient mice were lethally irradiated for total ablation of the native bone marrow and consecutively reconstituted with isolated bone marrow cells from WT donor mice. By injecting streptavidin-coupled CXCL1 intratracheally into these mice 5 weeks after successful bone marrow transplantation, we demonstrated that the loss of Btk in non-hematopoietic cells in the lung significantly reduces CXCL1 transcytosis from the alveolar to the vascular compartment in the lung (Fig. 9A). To investigate if CXCR2-mediated chemokine transcytosis involved chemokine-induced Gαi-mediated signaling, we pretreated WT mice with pertussis toxin (Ptx). Ptx treatment did not reduce CXCL1 transcytosis as compared to saline-treated control mice, indicating that CXCR2-mediated chemokine transcytosis is independent of convention chemokine signaling through this receptor (Fig. 9A). On a molecular level, the loss of Btk led to reduced phosphorylation of p38 and Akt in isolated alveolar epithelial cells after stimulation with KC (Fig. 9C). To demonstrate the functional relevance of BTK in vivo, WT and Btk-KO recipient mice were lethally irradiated for total ablation of the native bone marrow and consecutively reconstituted with isolated bone marrow cells from WT donor mice. After induction of bacterial pneumonia, the loss of non-hematopoietic Btk expression led to defective neutrophil recruitment into the lung and increased CFU counts (Fig. 10A-D).

## Discussion

It is well established that the engagement of CXCR2 on leukocytes is involved in the activation and recruitment of these cells, whereas the functions of this particular GPCR on different epithelial and endothelial-vascular beds within the lung parenchyma remain unclear. In this study, we demonstrate that the non-hematopoietic expression of CXCR2 on epithelial and endothelial cells in the lung is indispensable for efficient chemokine transcytosis of CXCL1 from the alveolar space through the lung interstitial space towards the apical side of endothelial cells in the lung microvasculature. This process is dependent on the presence of the tyrosine kinase Btk in these cells. If the process of CXCR2-mediated trans-epithelial and trans-endothelial CXCL1 transcytosis is disturbed, efficient neutrophil recruitment into the lung during the onset of bacterial-induced pulmonary inflammation is abolished, and bacterial clearance is severely impaired.

Among the major chemokine-binding GPCRs, CXCR2 stands up as a particularly interesting receptor because of its broad expression on many hematopoietic and non-hematopoietic cells[13–15]. Although primarily expressed on myeloid leukocytes, CXCR2 is found on all lung endothelial cells and on endothelial progenitors, as well as on subsets of lung epithelial cells[14–16]. In support of these findings, we could also demonstrate the expression of CXCR2 on epithelial and endothelial cells in the murine lung. Importantly, lung CXCR2 is constitutively expressed and does not require initial gene transcription and translational upregulation by strong inflammatory signals, which appears

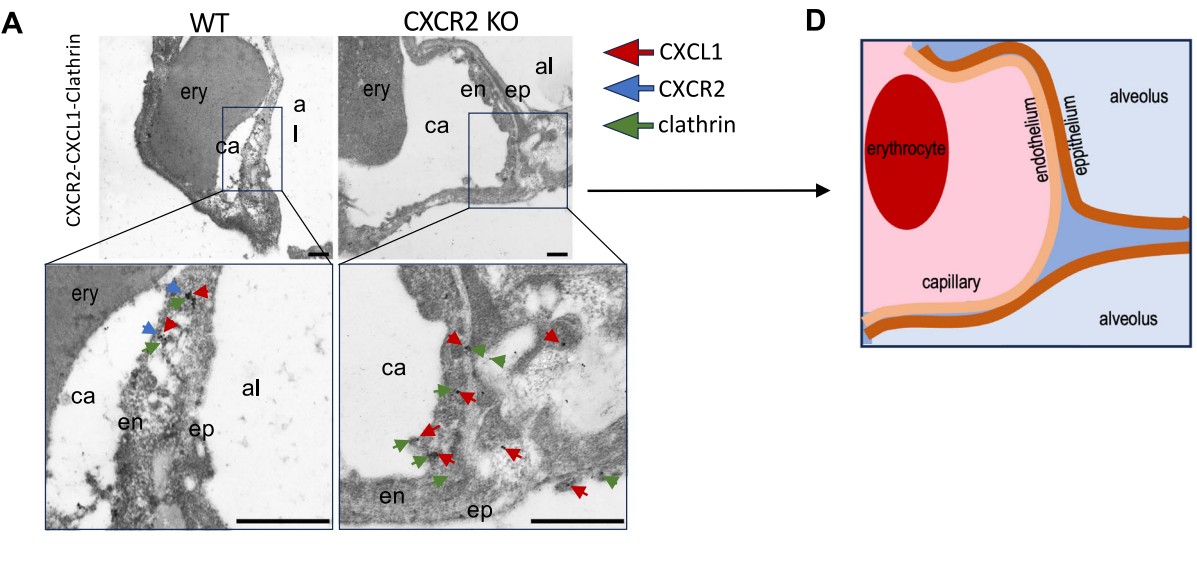

**Fig. 7 | Transmission electron microscopy of murine lung sections reveals evidence for a CXCR2-mediated transcytosis of CXCL1.** Colocalization of CXCR2 (blue), CXCL1 (red), and clathrin (green) was analyzed in ultrathin cross-sectioned lung tissue from *K. pneumoniae* infected CXCR2 KO and ctrl mice by transmission electron microscopy (TEM). Immunological staining was performed using gold particles of different sizes (6, 12, and 18 nm) coupled with antibodies against CXCR-2, CXCL-1, and clathrin. Lung sections from WT and CXCR2 full knockout mice (**A**) as well as conditional KO mice and corresponding Cre⁻ controls (**B**, **C**). Abbreviations: ery erythrocyte, ca capillary, en endothelium, ep epithelium, al alveolar space. Scale bar: 500 nm (**A**), 200 nm (**B**, **C**). **D** Schematic cartoon of the morphological structures observable in (**A**). Numbers of CXCL1-coupled gold particles on (**E**) lung epithelial (EP) and (**F**) lung endothelial cells (EC) (data are mean ± SD analysis of $n = 9$ representative images per group), one-way ANOVA followed by Bonferroni correction, *$p < 0.05$.

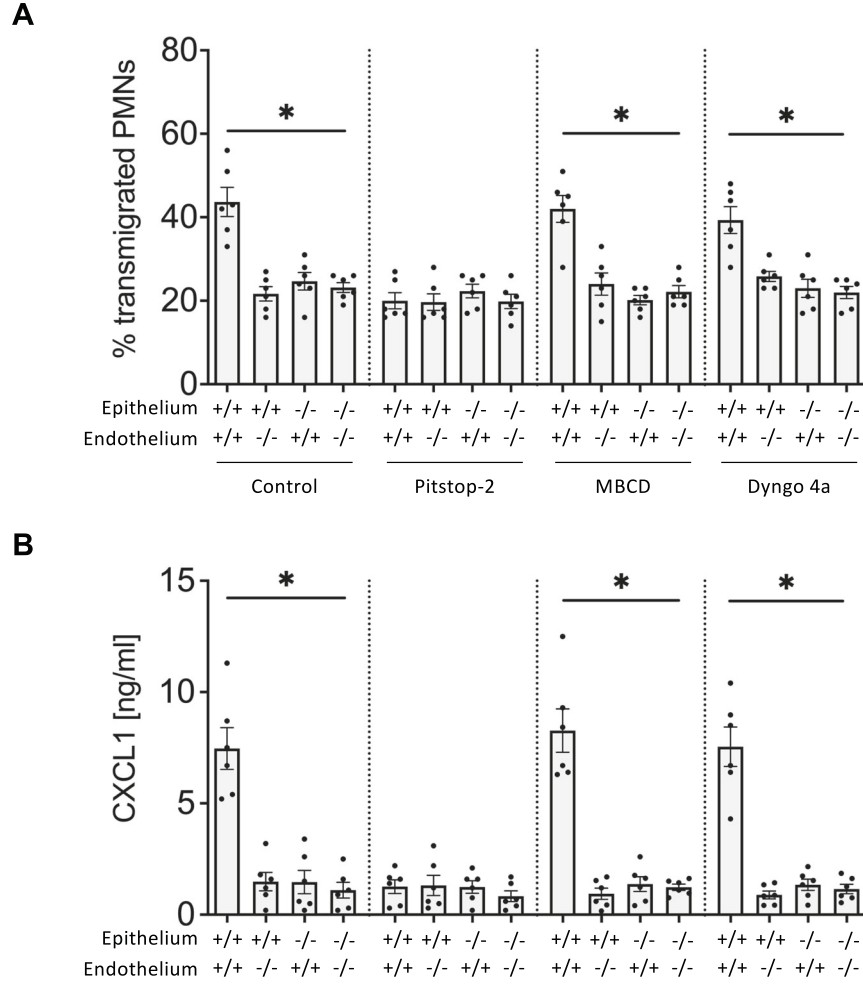

**Fig. 8 | Inhibition of clathrin-mediated transcytosis reduces CXCR2-mediated transcytosis of CXCL1. A** Primary murine lung microvascular endothelial cells (MLMVEC) and alveolar epithelial cells (AEC) were cultured on different sides of a Transwell™ insert. Transmigration of WT bone marrow-derived neutrophils towards a CXCL1 gradient was analyzed under control conditions (solvent DMSO) and after incubation of the epithelium and endothelium with the inhibitor Pitstop-2 (10 μM), MBCD (1 μM), and Dyngo 4a (30 μM) for 30 min. **B** CXCL1 concentrations in the upper well after 30 min (data are mean ± SD, $n = 6$ biologically independent samples), one-way ANOVA followed by Bonferroni correction, *$p < 0.05$.

consistent with the concept of specialized contribution for CXCR2 in acute inflammatory conditions in the lung parenchyma where immediate neutrophil recruitment would be of imminent importance. While the role of CXCR2 on hematopoietic cells has been extensively studied and characterized, less attention has been paid to the contribution of CXCR2 on non-hematopoietic cells on the course of infectious and inflammatory diseases. Yet, non-hematopoietic CXCR2 in the lung has been implicated in LPS induced changes in vascular permeability and in neutrophil recruitment to acutely inflamed lungs[23]. In this earlier study, LPS inhalation induced neutrophil recruitment to the lung much less efficiently in global CXCR2[−/−] mice reconstituted with WT bone marrow than in WT mice reconstituted with WT bone marrow[23]. The results of our study are in line with these observations, but unlike this earlier study, we could show here using cell-type specific conditional CXCR2 knockout mouse strains that CXCR2 expression on both epithelial and endothelial cells in the lung is necessary for chemokine transcytosis, leukocyte recruitment, and appropriate bacterial clearance in the lung. Interestingly, we found that the BALF protein content is elevated in global CXCR2[−/−] mice, but not in conditional endothelial or epithelial KO mice. This finding might be explained by the fact that additional CXCR2[+] cells, e.g., tissue resident mast or dendritic cells, are involved in the regulation of pulmonary

permeability but not in the process of CXCR2-mediated chemokine transcytosis.

Bacterial infection elicits rapid recruitment of neutrophils and inflammatory monocytes across the pulmonary capillaries into the alveolar space during the onset of pulmonary inflammation[25,26]. However, the very first response to alveolar infection is mediated by resident alveolar macrophages, the epithelial cells lining the alveoli, and subsets of interstitial dendritic cells which identify pathogenic bacteria and release multiple chemoattractants to rapidly recruit neutrophils[27–30]. The Atypical Chemokine Receptor 1 (ACKR1), formally known as Duffy Antigen receptor for Chemokines (DARC), serves a counter receptor for different CXC-motif chemokines, including CXCL1, and serves in chemokine transcytosis across both non lymphoid and lymphoid post capillary venules and binds CXCL1 with similar affinity as CXCR2[9,31–33]. Interestingly, normal lung capillary bed lacks expression of ACKR1[11,12], but it is expressed by the sparse small venules in the bronchial submucosa (see Supplementary Fig. 4D-E). We show ACKR1 protein expression by immunohistochemistry using a fully validated highly specific antibody. This is the best and only way to analyze the very sparse ACKR1 expression in the lung as even scRNA approaches miss ACKR1 signals because the population of ACKR1-positive cells is so rare. Furthermore, in mice (in contrast to humans or

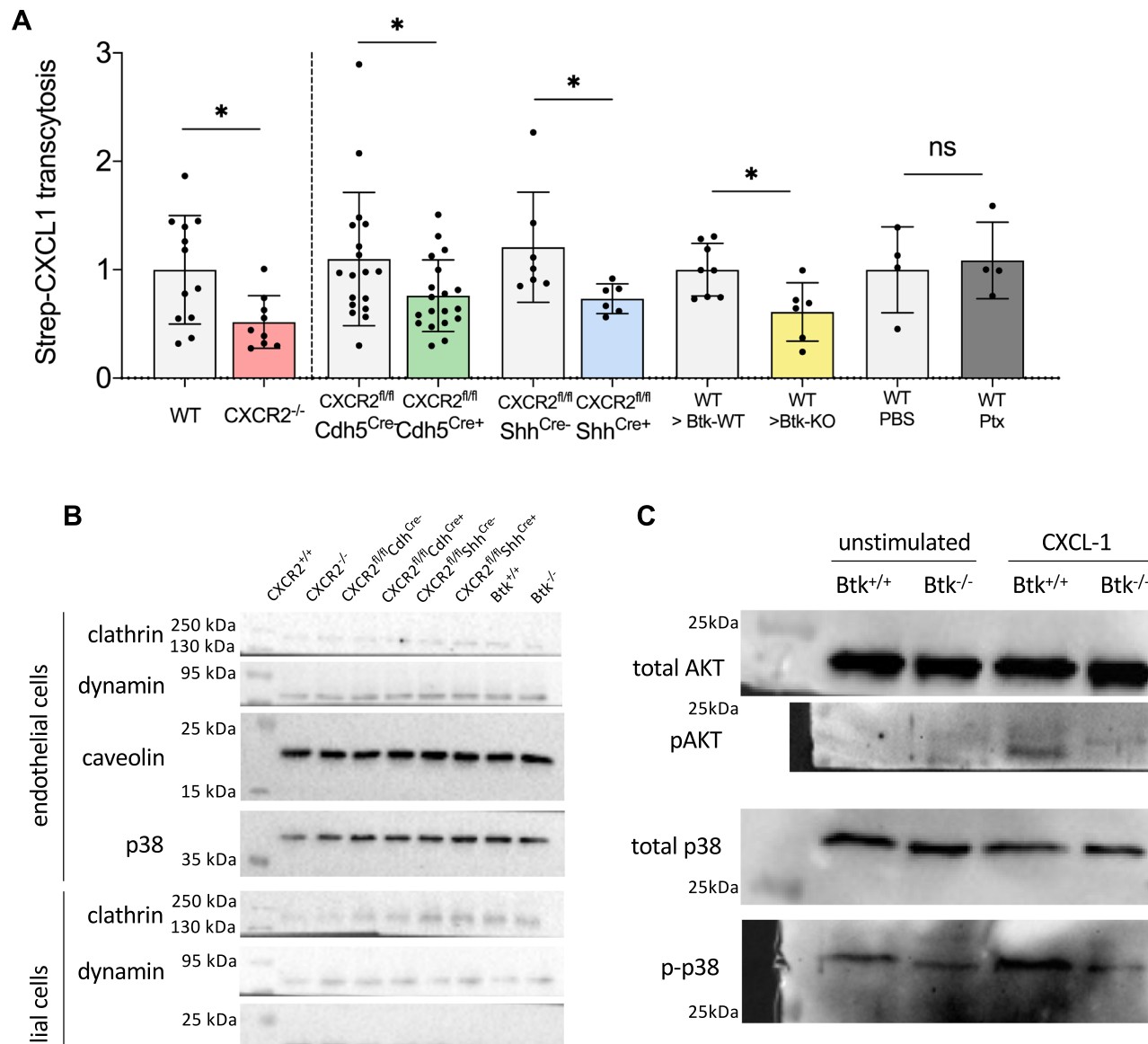

**Fig. 9 | Loss of endothelial and epithelial CXCR2 decreases transcytosis of CXCL1. A** CXCR2 KO mice (red=CXCR2$^{-/-}$, green=CXCR2$^{fl/fl}$Cdh5$^{Cre+}$, blue=CXCR2$^{fl/fl}$Shh$^{Cre+}$), appropriate WT/Cre$^-$ control (gray) mice, lethally irradiated WT or Btk$^{-/-}$ recipient mice reconstituted with isolated WT donor bone marrow, and WT mice pretreated with 4 μg Ptx/mouse i.v. or PBS as a control were injected intratracheally with 5 μg streptavidin-coupled CXCL1. After 4 h, blood samples were obtained and the levels of transcytosed streptavidin-coupled CXCL1 were analyzed by an ELISA using a CXCL1 capture antibody and a biotin-HRP conjugated detection antibody. Data are mean ± SD, $n = 4$–19 biologically independent mice pre group,

age 8–16 weeks, equal gender distribution, one-way ANOVA followed by Bonferroni correction, *$p < 0.05$. **B** Protein expression of clathrin (MW 190 kDa), dynamin (MW 100 kDa), caveolin (MW 24 kDa) and total p38 (MW 38 kDa) as protein loading control was analyzed by western blotting of pulmonary epithelial and endothelia cells isolated form the indicated mouse strains (exemplary images from $n = 3$ independent biological experiments). **C** Akt and p38 phosphorylation in alveolar epithelial cells isolated from Btk$^{+/+}$ and Btk$^{-/-}$ mice (exemplary images from $n = 3$ independent biological experiments), one-way ANOVA followed by Bonferroni correction, *$p < 0.05$.

larger mammals) the bronchial circulation supplies only a miniscule proportion of the pulmonary tissue (the trachea and main bronchi) while the majority of the lung parenchyma is supplied only by the pulmonary circulation[34]. However, in chronic lung inflammatory diseases, ACKR1 expression is induced in broader segments of vessels lining the pulmonary bronchiolar tree, possibly also in vessel directly lining the alveolae[35]. This suggested that in the alveoli ACKR1-independent transport mechanisms, and chemokine receptors might

be involved in chemokine transcytosis across the alveolar epithelial and endothelial barriers to substitute missing ACKR1[10,36,37]. Our data in this study suggest that CXCR2 does not function as transcytosis receptor in non-pulmonary organs, including the cremaster muscle and the injured kidneys. Why CXCR2 in the lung would be privileged to transcytose CXCR2 over other receptors, e.g., ACKR1, remains elusive. One aspect may be the constant exposure of the large internal lung surface to air-borne pathogens, requiring a very efficient fist line of

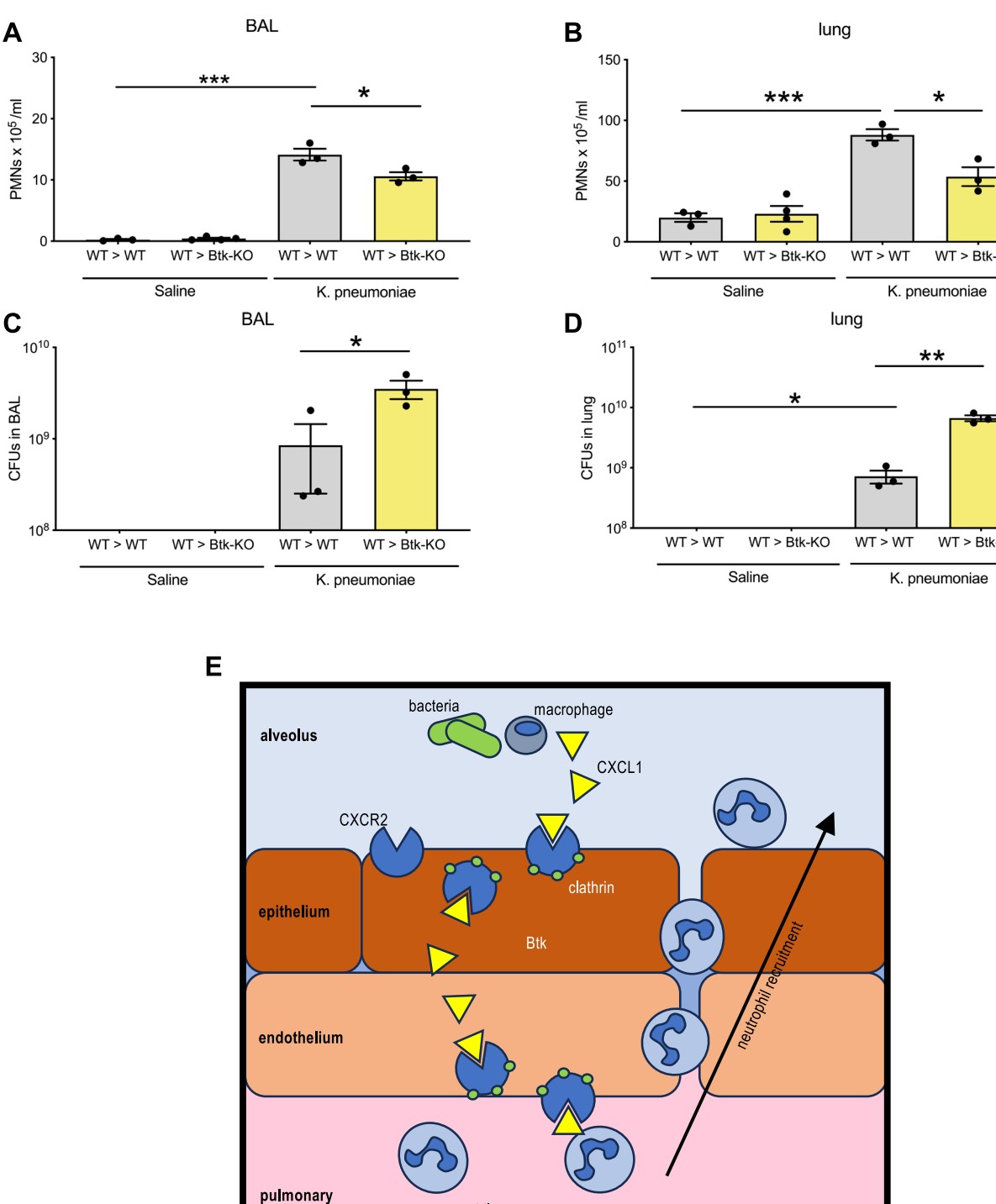

**Fig. 10 | Loss of non-hematopoietic Btk decreases transcytosis of CXCL1.** WT and Btk⁻/⁻ mice were leathally irradiated and bone marrow reconstitution was performed by i.v. injection of isolated WT or Btk⁻/⁻ bone marrow cells. 6 weeks after irradiation, bone marrow chimeric mice (WT donor -> WT recipient and WT donor -> Btk KO recipient) mice were intratracheally administered with sterile saline or with viable *K. pneumoniae*. Neutrophil recruitment into the BAL (**A**) and lung (**B**) as well as CFUs in the BAL (**C**) and lung (**D**) were assessed 24 h after infection. **E** Graphical abstract visualizing the main findings of this study. Data are mean ± SD, $n = 3$ biologically independent mice pre group, age 8–16 weeks, random gender distribution, one-way ANOVA followed by Bonferroni correction, $*p < 0.05$, $**p < 0.01$, $***p < 0.001$.

innate immune defense, but more detailed investigations may be required to address this specific question. In other vascular beds classical chemokine GPCRs have been implicated, albeit not unequivocally shown to be involved, in chemokine transcytosis, e.g., CXCR4 in bone marrow sinusoids[38] and CCR2 in brain vasculature[39]. Interestingly, transcytosis mechanisms have also been described not only for chemokines but other leukocyte chemoattractants, such as the complement component C5a where it has been previously demonstrated that atypical complement C5a receptor 2 (C5aR2) and ACKR1 expressed on endothelial cells were required for the transport of C5a and CXCR2 ligands into the vessel lumen in a murine model of immune complex–induced arthritis[33]. These observations are in line with our

findings and indicate that chemokine transcytosis appears to be an essential mechanism, although reflecting chemoattractant and tissues-specific molecular traits. In the lung, chemokines have to be transported through two distinct barriers, i.e., the epithelium and the endothelium, in order to be presented at the luminal side of pulmonary blood vessels. We here describe that CXCR2-mediated CXCL1 transcytosis is important in both of these cell types, which to our knowledge, has not been described previously.

We here demonstrate that in the lung, CXCR2 on pulmonary epithelial and endothelial cells mediates transcytosis of CXCL1 towards the luminal side of the pulmonary microvasculature to activate and recruit neutrophils. Of note, we show that lung epithelial and endothelial cells appear to produce low baseline amounts of endogenous CXCL1 located intracellularly themselves. This is in line with previous reports indicating that these endogenous CXCL1 levels under baseline conditions do not induce immune cell recruitment, either because the concentrations are too low or the intracellular stored chemokines are less likely to be mobilized and presented on the luminal side of lung endothelial cells[40]. We also demonstrate that administration of a relatively small exogenous amount of the chemokine CXCL1 is sufficient to trigger the immediate recruitment of circulating neutrophils into the non-infected "resting" lung parenchyma. Interestingly, CXCL1 levels in the circulation were found to be increased in CXCR2-deficient mice, resembling an overshooting immune system activation in the circulation, which is also indicated by the increased CFU counts in the blood of these mice. This indicates that early neutrophil recruitment into the lung, mediated by a relatively small amount of transcytosed CXCL1 originating from the alveoli, is necessary to efficiently contain bacterial infections and subsequent pathogen dissemination into the circulation. However, unlike CXCR2, CXCL1 binding to ACKR1 does not induce neither G-protein-dependent nor independent signal transduction. Analogously, it is possible that in the absence of appropriate intracellular secondary signaling effectors CXCR2 in both epithelial and endothelial cells is also kept as a "silent" GPCR, similar to ACKR1 allowing it to mediate transcytosis. This concept of biased GPCR signaling independent of G proteins has gained a broader attention lately[41,42]. In line with this concept, C5aR2 on endothelial cells bind C5a but recycles without being degraded[43]. C5aR2 has been shown to elicit cellular signaling via ß-arrestin[44]. Interestingly, GPCRs bound to ß-arrestin recruit PTKs to specialized assemblies. Classical ACKRs also bind ß-arrestin in a G-protein-dependent but ß-arrestin-independent manner[41]. These processes may also be regulated by ERK. Efficient CXCR2-mediated transcytosis requires Btk a non-receptor tyrosine kinase expressed in hematopoietic and non-hematopoietic cells. In line with these findings, previous reports showed that Btk in non-hematopoietic pulmonary tissue is required for neutrophilic inflammation in a murine model of asthma[45]. Apart from the lung, non-hematopoietic Btk has also been shown to aggravate infectious inflammation in the kidney[46]. In contrast to both aforementioned reports, however, this study is to our knowledge the first to report a role of non-hematopoietic Btk on CXCR2-mediated chemokine transcytosis in the lung. Both p38 MAPK and Akt signaling pathways may regulate clathrin-coated pit (CCP)-mediated transcytosis by modulating vesicle formation, cargo recruitment, and cytoskeletal dynamics. p38 MAPK enhances dynamin activity, facilitating vesicle budding, and regulates actin cytoskeleton remodeling via phosphorylation of heat shock protein 27 (HSP27), which supports actin-dependent vesicle movement[47,48]. Additionally, p38 influences Rab GTPases involved in vesicle sorting and maturation after endocytosis. Akt signaling contributes by regulating adapter proteins such as AP-2, which are essential for cargo selection and clathrin recruitment, and by modulating Rab proteins like Rab11 to direct vesicles through the transcytotic pathway[49]. Together, p38 and Akt may coordinate the internalization, trafficking, and exocytosis of CXCL1, with p38 supporting vesicle scission and actin remodeling, while Akt

enhances cargo selection, sorting, and vesicle fusion, ensuring efficient CXCL1 transcytosis.

Receptor-mediated transcytosis (as suggested here for CXCL1 via CXCR2) requires changes in receptor affinity as a function of the transcytosis process to ensure that the ligand is taken up with high affinity on the one side but released with a lower receptor affinity on the other side of the lung interstitium. However, the investigation of the molecular mechanisms guiding these processes is beyond the scope of this study and warrants further research.

As a limitation, we have only analyzed neutrophil recruitment and bacterial CFU counts 24 h after infection. This time point is commonly used in studies focusing on the investigation of specific molecular mechanisms involved in the regulation of leukocyte activation and recruitment[50–52], and the infection course and bacterial invasion behavior of the bacteria itself was not the focus of this study.

Taken together, our data demonstrate that the expression of CXCR2 on lung endothelial and epithelial cells drives CXCL1 transcytosis through pulmonary endothelial and epithelial cells in a Btk-dependent manner. The cell type-specific genetic deletion of CXCR2 in mice leads to impaired neutrophil recruitment and host defense against invading pathogens during bacterial-induced pneumonia. Furthermore, this role of CXCR2 on non-hematopoietic cells in the lung appears to be organ-specific and non-systemic. Indeed, lung targeted CXCR2 blockade by intratracheal injection of blocking antibodies led to worsening of bacterial dissemination in a model of bacterial-induced pulmonary inflammation. Conversely, targeting CXCR2-mediated chemokine transcytosis might be a viable potential strategy of therapeutic intervention in non-infectious inflammatory pathologies in the lung, a possibility that warrants further research.

## Methods

### Mice
Male and female WT (Charles River), CXCR2$^{-/-}$ (Jackson Lab, stock 006848), CXCR2$^{fl/fl}$Cdh5$^{Cre}$ (Taconic stock 13073) and CXCR2$^{fl/fl}$Shh$^{Cre}$ mice (Jackson Lab, stock 005623) on a C57BL/6 background were kept in a specific pathogen-free/SPF barrier facility with a 12:12 light:dark cycle and food and water ad libitum. Experimental and control animals were co-housed. According to the Jackson Laboratory guideline for the intraperitoneal injection of tamoxifen for inducible cre-driver lines, CXCR2$^{fl/fl}$Cdh5$^{Cre}$ mice were treated with tamoxifen (2 mg tamoxifen in 100 µl corn oil) once every 24 h for a total of 5 consecutive days via intraperitoneal injection. All mouse experiments were approved by the North Rhine-Westphalia Office for Nature, Environment and Consumer Protection ("Landesamt für Natur-, Umwelt- und Verbraucherschutz NRW"; reference number 81-02.04.2019.A445). Specific termination criteria were specified in the animal protocol (abnormal pain, abnormal behavior, high morbidity) and adhered to throughout the experiments.

### Bacteria-induced pulmonary infection
Age and sex-matched mice were infected with viable *Klebsiella pneumoniae* or *Escherichia coli*. Neutrophil recruitment in the lung was analyzed after 24 h as described previously[50,53]. In brief, overnight cultures (37 °C, 250 rpm) of *K. pneumoniae* (ATCC strain 13883) or *E. coli* (ATCC strain 25922) were grown in tryptic soy broth, washed and resuspended in sterile PBS. Mice were anaesthetized by intraperitoneal injection of ketamine (125 µg/g body weight, "Wirtschaftsgenossenschaft Deutscher Tierärze, WDT") and xylazine (12.5 µg/g body weight; Bayer) and challenged with $4 \times 10^7$ *K. pneumoniae* or $6 \times 10^6$ *E. coli* per mouse in 50 µl sterile PBS via intratracheal injection. For some experiments, a CXCR2 blocking antibody (30 µg/mouse; R&D Systems, #MAB2164) was injected simultaneously with the intratracheal installation of *K. pneumoniae*. Mice were euthanized by anesthesia overdosing and blood was drawn via cardiac puncture. The lung was lavaged four times with 0.7 ml of a physiologic saline solution

(NaCl 0.9%, B. Braun) and the lung circulation was perfused with 3 ml PBS. Lungs were enzymatically digested with DNase I, collagenase type XI, and hyaluronidase type I-s (Merck, #D4527, # C7657, # H3506). The number of recruited neutrophils in the BAL and the lung tissue was determined via Kimura staining and following CD45/Ly-6B.2/Gr-1 (CD45-PerCP/Cy5.5 (clone 30/F11, BioLegend, #103132), Ly-6B.2-FITC (clone 7/4, Bio-Rad, #MCA771), Gr-1-AF633 (clone RB6-8C5, purified from hybridoma supernatant) antibody staining for flow cytometry-based analyses. Colony forming unites (CFU) in the BAL, lung, blood, and spleen were counted by serial plating on tryptic soy agar plates.

### Intravital microscopy of the lung

Intravital microscopy (IVM) of the murine lung was performed using a thoracic suction window as described previously, with some modification[53]. Briefly, mice were anaesthetized by intraperitoneal injection of ketamine (125 µg/g body weight, WDT) and xylazine (12.5 µg/g body weight; Bayer). An endotracheal tube was placed directly into the trachea via a tracheotomy, and the mice were ventilated mechanically with a MiniVent Type 845 small rodent ventilator (Harvard Apparatus, tidal volume 10 µl/g bodyweight, respiratory frequency 150/min, PEEP 5cmH$_2$O). A thoracotomy was performed to expose the right middle lung lobe and the lung was hold in position using a custom-built fixation device with an integrated observation window. To visualize neutrophil recruitment, mice were intravenously injected with an AF488-coupled anti-Gr-1 antibody (clone RB6-8C5, purified from hybridoma supernatant) 4 h before imaging. Simultaneously, CXCL1 (5 µg, Peprotech, #250-11) or fMLP (9 µg, Sigma, #F3506) was intratracheally injected. Neutrophil recruitment into the BALF was quantified by flow cytometry 4 h later. The intravital microscopy by fluorescence microscopy does not allow the precise allocation of the neutrophils within the different compartments of the lung. Therefore, we analyzed the presence of interacting neutrophils in the lung as cells remaining stationary for >30 s in the pulmonary vasculature, the interstitial space or the alveoli. The number of cells was counted per field of view (FOV).

### Confocal lung imaging ex vivo

The analysis of viable lung sections was performed as described before[50,54]. Briefly, mice were injected i.t. with CXCL1. After 4 h, animals were injected with Alexa Fluor 488–coupled anti-Ly6G antibody (clone 1A8; 5 µg/mouse, BioLegend, #127626) and Alexa Fluor 568–coupled anti-PECAM antibody (clone 390; 50 µg/mouse; BD Biosciences, #558736) to stain neutrophils and endothelial cells. Mice were euthanized by anesthesia overdosing, and lungs were filled with 1 ml of low-melting agarose. After removal, lungs were cut using a vibratome. Lungs were fixed in a cell culture dish and submersed in PBS, and z-stacks were recorded using a spinning disc confocal microscope (CellObserver SD; Carl Zeiss) equipped with a 20×/1.0 NA objective.

### Isolation of murine lung microvascular endothelial cells (MLMVEC)

Mice were euthanized by cervical dislocation, the lung lobes were removed and mechanically minced. The tissue homogenisates was enzymatically digested with collagenase A (Roche, # 0103578001) and DNAse 1 (Merck, #D4527) at 37 °C for 90 min with gentle agitation before passing the cell suspension over a 70 µm cell strainer and centrifugation at 300 x g for 5 min. 20 µl magnetic dynabeads (Invitrogen) were incubated with 15 µg anti-CD31 antibody (clone Mec13.3, BD Biosciences, #553369) overnight at 4 °C on a shaker. After washing, the dynabeads were incubated with the cell suspension for 45 min at 4 °C on a shaker. The dynabead-bound cells were passed six times over a magnetic separator (Milltenyi), resuspended in DMEM medium (supplemented with 20% FCS), and seeded on gelatin-coated cell culture dishes. Cells were kept in an cell culture incubator at 37 °C and 10%

CO$_2$. Dynabead-assisted cell sorting for purification was repeated twice per week for a total of 4-5 purification cycles.

### Isolation of alveolar epithelial cells (AEC)

After euthanasia the thorax was opened and the trachea was intubated with a 20 G plastic canula. The lung was filled with 1 ml Dispase solution, followed by 0.5 ml of a 1% low-melt agarose (Promega, #V2111) heated to 45 °C. After 2 min of external cooling of the lung, the lung lobes are removed, placed into 0.5 ml of Dispase and incubated at 25 °C for 45 min with gentle agitation. Afterwards, the lung was minced into small pieces and serially passed over cell strainers with 40 and 20 µm pore size. The cell suspension was centrifuged at 200 x g for 7 min and incubated in erythrocyte lysis buffer supplemented with 10 µl DNAse I (Merck, #D4527) for 2 min before centrifugation at 300 x g für 5 min. The pellet was resuspended in DMEM supplemented with 10% FCS, 1% Penicillin/Streptomycin and 1% Glutamine. Subsequently, negative selection was performed with 1 µg of biotinylated CD45 antibody (clone 30-F11, BD Biosciences, #553076) and 0.65 µg of biotinylated CD16/32 antibody (clone 2.4G2, BD Biosciences, #567021) per $1 \times 10^6$ cells and incubated with 100 µl washed MagneSpheres (Promega, #Z5481) for 30 min. The cell suspension was passed three times over a magnetic separator for 3 min and the supernatant was collected. The cells were seeded on cell culture dishes coated with 15 µg/ml fibronectin (Merck, #F0895) and incubated at 37 °C and 5% CO$_2$.

### In vitro transmigration assay

Transmigration of bone marrow-derived neutrophils was analyzed in an in vitro bilayer model. Therefore, murine lung microvascular endothelial cells (MLMVEC) and alveolar epithelial cells (AEC) were isolated from 15 to 20-week-old mice. For an alveolus-like vascular-to-alveolar orientation AECs were transferred onto the bottom side of a Transwell™ insert (5 µm pores, Corning, #CLS3421) and allowed to adhere for 12 h before MLMVECs were seeded onto the top side of the Transwell™. Cells were allowed to grow to confluence (usually 5 days), and confluency was verified via dextran blue (20 kDa, Merck, #03714p) exclusion. Bone marrow-derived neutrophils were isolated via Pancoll density centrifugation and $4.5 \times 10^5$ cells were added into the upper chamber of the Transwell™. Neutrophils were allowed to migrate through the bilayer for 3 h following a CXCL1 gradient (100 ng/ml, Merck, #250-11) applied to the lower Transwell™ chamber. The number of transmigrated neutrophils was determined by the help of a Sysmex hematology analyser.

### Transmission electron microscopy

For fixation, mice were perfused with 2% (v/v) formaldehyde and 0.25% (v/v) glutaraldehyde in 100 mM cacodylate buffer, pH 7.4, at 4 °C. After fixation for 2 h, samples were cut in small pieces and rinsed in PBS. After dehydration in ethanol up to 70%, samples were embedded in LR White embedding medium (London Resin Company, London, UK) according to the manufacturer´s instructions using beem capsules. The embedding medium was polymerized using UV light. Ultrathin sections were cut with an ultramicrotome and collected on copper grids. For immunogold electron microscopy, ultrathin sections were incubated with 100 mM glycin in PBS for 2 min, washed with PBS and blocked for 30 min with 2% (w/v) BSA and 1% normal goat serum (Aurion) in PBS. Afterwards, ultrathin sections were incubated for 1 h at room temperature on drops of primary rat antibodies against CXCR-2 (R&D Systems, # MAB2164) diluted 1:50, primary rabbit antibodies against CXCL-1 (Invitrogen, #MA5-23745) diluted 1:25, and primary goat antibodies against clathrin diluted 1: 100 in PBS containing 1% (v/v) BSA-c (Aurion) and 0.025% (v/v) Tween 20. After washing with the same solution, ultrathin sections were incubated with secondary antibodies conjugated to 18 nm, 12 nm and 6 nm gold particles. After washing with distilled water, ultrathin sections were negatively stained

with 2% (w/v) uranyl acetate for 15 min. Electron micrographs were taken at 60 kV with a Philips EM-410 electron microscope using imaging plates (Ditabis, Pforzheim, Germany). Single electron microscopic images were analyzed for distribution of CXCL1 (12 nm sized gold particles) on endothelial and epithelial cells in a double-blinded manner by multiple researchers.

## Strep-CXCL1 injection and ELISA

Murine recombinant CXCL1 was labeled with streptavidin by using the Streptavidin Conjugation Kit (Abcam, #ab102921) according to the manufacturer's instructions. Streptavidin labeled CXCL1 (5 μg) was intratracheally injected, and plasma was collected after 4 h. To determine streptavidin-CXCL1 levels in the plasma an ELISA plate was coated with a CXCL1 capture antibody (CXCL1 mouse DuoSet ELISA Kit, R&D, #DY453) over night. Wells were washed three times with washing buffer and coated with reagent diluent (R&D) for at least 1 h. Wells were washed and 200 μl plasma sample were applied per well. After an incubation of 2 h, wells were washed and a biotinylated peroxidase (1:10,000 in reagent diluent) was added for 30 min. Wells were washed again before a TMB substrate (BioLegend, #421101) was used to provoke a color change. Sulfuric acid was supplemented to terminate the reaction. The ELISA plates were quantitatively analyzed at 450 and 540 nm in a Synergy 2 (BioTek/Agilent) plate reader.

## CXCR2 expression analysis via flow cytometry

Flow cytometry was performed to assess CXCR2 expression on murine, pulmonary epithelial and endothelial cells. Mice were euthanized by cervical dislocation and the lungs perfused by flushing 3 mL PBS through the right ventricle of the heart. Lungs were removed, minced, digested and homogenized. For CXCR2 expression analysis (clone SA044G4, BioLegend, #149302) lung homogenates were stained based upon the expression of CD45⁻ (clone 30/F11, BioLegend, #103132) CD31⁺ (clone 390, BD, #558736) for endothelial cells, or CD31⁻ EpCAM⁺ (clone G8.8, BioLegend, #118202) for epithelial cells. Samples were analyzed in a FACSCantoII Flow Cytometer (BD Bioscience).

## Surface marker expression on pulmonary endothelial and epithelial cells

Isolated MLMVECs and AECs were analyzed for the expression of classical endothelial and epithelial surface markers. MLMVECs and AECs were isolated as mentioned previously and stained with specific fluorescent antibodies and appropriate isotype controls. MLMVECs were stained for CD31 (clone MEC13.3, BioLegend, #553369), CD54 (clone YN1/1.7.4, BioLegend, #116102), CD102 (clone 3C4, BioLegend, #105602), CD321 (clone H202-106, BD Bioscience, #744772), CD323 (clone 206928, R&D, #AF1213), TLR2 (clone CB225, BioLegend, #148601) and TLR4 (clone MTS510, Invitrogen, #14-9924-82). While expression of CD326 (clone G8.8, BioLegend, #118202), CD54 (clone YN1/1.7.4, BioLegend, #116102), CFTR (clone CF3, Abcam, #ab2784), Muc-1 (polyclonal, Abcam), SP-C (polyclonal, Abcam, #ab90716), TLR2 (clone CB225, BioLegend, #148601) and TLR4 (clone MTS510, Invitrogen, #MABF2274) was determined on AECs two and seven days after isolation. Samples were analyzed in a FACSCantoII Flow Cytometer (BD Bioscience).

## Immunofluorescence staining

Mice were killed, lungs removed and snap-frozen in OCT on dry ice and stored at -80°C. 7 μm thin sections were cut on a cryo-microtome, fixed in 4% PFA for 5 minutes, then washed in 2 changes of PBS. The sections were blocked with 10% goat serum (Merck, #G9023) in PBS/BSA (0.5% BSA in PBS) for 30 min, followed by staining for 30 min with purified unconjugated anti-ACKR1 (clone 6B7, produced in-house) diluted 1:500 in PBS/BSA. After 3x washes sections were stained with a secondary goat anti-mouse antibody conjugated with AF488 (Invitrogen (ThermoFisher

Scientific, Waltham, MA, USA) diluted 1:500 in PBS/BSA and the nuclei counterstained with DAPI (ThermoFisher Scientific, Waltham, MA, USA) for 5 min. After three further washes the slides were mounted with Prolong Gold (ThermoFisher Scientific, Waltham, MA, USA) and left to cure overnight at room temperature. Images were taken on the SM800 confocal microscope (Zeiss, Oberkochen, Germany).

## CXCL1 ELISA and BCA assay

Plasma and BAL samples from mice that underwent the lung infection procedure were collected to assess cytokine and total protein concentrations. CXCL1 was determined by using the Mouse CXCL1/KC Quantikine ELISA Kit (R&D Systems, #MKC00B) according to the manufacturer's instructions. Likewise, the total protein content in the BAL was investigated by the help of the Pierce™ BCA protein assay kit (Thermo, #23227). The ELISA plates were quantitatively analyzed in a Synergy 2 (BioTek/Agilent) plate reader.

## Intravital microscopy of the cremaster muscle

Intravital microscopy of the murine cremaster muscle was performed as previously described[55,56]. Briefly, mice received either an intrascrotal injection of 500 ng TNF (R&D Systems, #410-MT-010) in 0.3 ml saline, CXCL1 superfusion 2 h or fMLP (10 μM) superfusion for 5 min[57] before cremaster muscle exteriorization. High-speed multichannel fluorescence microscopy was performed on an upright microscope (Axioskop; Carl Zeiss, Göttingen, Germany) equipped with a Lambda DG-4 ultra-high speed wavelength switcher (Sutter Instruments, Novato, CA, USA) and a 40 × 0.75 NA saline immersion objective. Videos were recorded with a digital camera (Sensicam QE) and analyzed with Slidebook Software (Version 5; Intelligent Imaging Innovations, Göttingen, Germany). Transmigration was analyzed using the reflected light oblique transillumination microscopy (RLOT)[58].

## Renal ischemia/reperfusion injury (IRI)

The IRI model has been described previously[59]. In brief, mice were anesthetized by intraperitoneal administration of ketamine (125 μg/g body weight, WDT) and xylazine (12.5 μg/g body weight; Bayer) and were placed on a heating pad to maintain body temperature. In animals undergoing IRI, both renal pedicles were clamped off for 35 minutes with hemostatic micro clips. After clamp removal, kidneys were checked for a change in color to ensure reperfusion. Incisions were closed in two layers. Animals were kept on a heating pad to maintain body temperature and had free access to food and water. Mice were euthanized by cervical dislocation 24 h later to investigate neutrophil recruitment into the kidney and plasma creatinine values were determined. Therefore, mice were perfused, the kidneys were removed, mechanically minced and enzymatically digested. Antibody staining and flow cytometric analysis was performed as described earlier to identify the number of recruited neutrophils. Plasma creatinine was assessed by using a creatinine assay kit (Diazyme, Poway, #DZ072B) according to the manufacturer's protocol.

## Western blotting

Isolated AECs or MLMVECs were either stimulated with CXCL-1 (100 ng/ml, 37°C, 10 mins) or left unstimulated. Following stimulation, cells were lysed in phospho lysis buffer while incubation on a shaker for 30 min at 4 °C. Cells were centrifuged at maximum speed for 15 min at 4 °C, and proteinlysates were boiled with sample buffer for 10 min at 95 °C. Samples were separated with 10% SDS-PAGE and transferred to a nitrocellulose membrane. Primary Akt (total and phosphor, #4691S and #4058S), p38 (total and phosphor, #9212S and #9214S), clathrin (#4796S), dynamin (#2342S), and caveolin antibodies (#3238S, all from Cell Signaling Technology) were incubated at 4 °C overnight. Secondary anti-rabbit IgG coupled to HRP (Cell Signaling Technology) was incubated for 1 h at room temperature. Immunoblots were developed using an ECL system (GE Healthcare).

## In vitro migration assay

In vitro migration assay was performed as described previously[51]. Murine neutrophils were seeded on fibronectin-coated (50 μg/ml) chemotaxis μ-slides (Ibidi). Within the chemotaxis slide, a CXCL-1 gradient was applied by diffusion of a Patent Blue (Merck, #198218) colored CXCL-1 solution (1 μg/ml, Peprotech) in one reservoir of the slide according to the manufacturer´s instructions. Cell movement was recorded with a microscope platform (37 °C, 5% $CO_2$, Axio Observer, Zeiss) over a period of 30 min by using time-lapse microscopy (3 frames/min). For analysis, cells were tracked with Manual Tracking ImageJ software (version v.1.54 h) and analyzed with Chemotaxis plug-in (Ibidi). Euclidean distance and forward migration index of the cells were analyzed.

## Statistics

All statistical calculations were performed using the GraphPad PRISM software (version 9). Data distribution was assessed using Kolmogorov-Smirnov-test or Shapiro-Wilks test, differences between groups were analyzed using the Wilcoxon-test or t-test as appropriate. More than two groups were compared using one-way ANOVA followed by Bonferroni correction. All data are represented in means ± SD. A $p$-value < 0.05 was considered statistically significant.

## Reporting summary

Further information on research design is available in the Nature Portfolio Reporting Summary linked to this article.

## Data availability

All data are included in the Supplementary Information or available from the authors, as are unique reagents used in this Article. The raw numbers for charts and graphs are available in the Source Data file whenever possible. Source data are provided with this paper.

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

## Acknowledgements

This work was funded by the Deutsche Forschungsgemeinschaft (DFG grant number ZA428/24-1, CO2096/2-1, ZA428/18-2, INST211/1073-1, ZA428/14-2, INST 211/984-1, INST211/604-3, TRR332 project C01 and CRC1450 project B05 to A.Z., RO 4537/4-2, RO 4537/5-2, and CRC1450 project C07 to J.R.), the IZKF Münster (Za2/001/18 to A.Z.) and GIF grant number I-1470-412.13/2018 (to R.A. and A.Z.). A.R. and E.H. were supported by the Wellcome Trust (Investigator Award 200817/Z/16/Z to A.R.).

## Author contributions

K.T. and N.L. conducted experiments, performed analysis of the data and helped writing the manuscript. S.M., N.K, J.G., L.S., M.O. and K.H. performed experiments. A.M., H.B., U.H., K.G., M.E., E.H. and J.E. performed experiments, analyzed and interpreted the data. K.H., V.M., V.G., J.Rot, A.R. and R.A. aided in experimental design and reviewed the manuscript. J.Ros. and A.Z. conceived of the study, conducted experiments, analyzed the data, supervised the study, and wrote the manuscript.

## Funding

## Competing interests

The authors declare no competing interests.
