## [Transparent Peer Review file · Nature Communications]

Alveolar epithelial and vascular CXCR2 mediates transcytosis of CXCL1 in inflamed lungs

Corresponding Author: Professor Alexander Zarbock

Version 0:

Reviewer comments:

Reviewer #1

(Remarks to the Author)

The study under review investigates the role of CXCR2 in neutrophil recruitment and bacterial clearance in the lung, utilizing global and cell-specific conditional KO mice. The findings suggest that CXCR2 deficiency hampers neutrophil recruitment and increases bacterial burden in all KO strains, with CXCR2 neutralization reducing neutrophil recruitment. The authors also report that pulmonary endothelial and epithelial CXCR2 support neutrophil recruitment, and KO mice exhibit decreased neutrophil transmigration upon CXCL1 stimulation. Using in vitro studies with a Transwell™ model, reduced neutrophil transmigration through endothelial-epithelial bilayers lacking CXCR2 is demonstrated. Additionally, transmission electron microscopy reveals CXCL1 clustering with CXCR2 in lung epithelial and endothelial cells during infection. In vivo experiments further show decreased CXCL1 transcytosis in CXCR2-deficient mice, implicating the involvement of Bruton's tyrosine kinase. The authors conclude that CXCR2 plays a role in alveolar epithelial and endothelial cells by mediating cognate CXCL1 transcytosis, favoring neutrophil recruitment to infected lungs.

While the study addresses an important topic, it predominantly relies on knockout mouse models, and there are notable gaps in experimental controls. The rationale behind the experimental design and the mechanistic pathway proposed require further elucidation. Below are detailed comments for consideration:

1. The manuscript would benefit from a more comprehensive characterization of the global and cell-specific CXCR2 knockout mice. The absence of protein expression data, relative to wild-type mice, and the lack of references for validation are notable omissions. Given the potential for compensatory genetic responses, additional validation experiments using blocking antibodies (or siRNA) against CXCR2 are recommended to substantiate the findings in these knockout mice.
2. The assertion that conditional CXCR2 knockout in endothelial and epithelial cells does not impact CXCR2 expression in circulating leukocytes warrants supporting evidence. The data should be included as supplementary material.
3. The study assesses ACKR1 expression solely via immunostaining, without a clear rationale for its relevance to the study's objectives. Further clarification on the reasoning behind this measurement and additional data to confirm the absence of this protein is necessary.
4. The lung infection model lacks critical negative controls, such as wild-type, Cre-positive, and Cre-negative control mice treated with phosphate-buffered saline, the solvent for intratracheal bacterial infusion. Conducting statistical analysis across all four groups post-experiment and replicating the experiments with blocking antibodies would strengthen the study's validity.
5. Discrepancies between Figure 1, which depicts two mouse groups treated with *K. pneumoniae*, and supplementary Figure 4, which includes four groups, raise questions. Clarification on whether the baseline and post-infection experiments were conducted concurrently would be beneficial.
6. Inconsistencies in the number of mice evaluated within the same group, as evidenced by varying data points across figures, necessitate an explanation or justification for any data exclusion. For example, in the CXCR2^{fl/fl}Cdh5Cre⁺ group, the number of dot plots varies between Fig 1D (n=7), Fig 1E (n=9), and Fig 1F (n=13).
7. The administration route for the CXCR2 blocking antibody, used concurrently with bacterial infusion, requires clarification. If administered intratracheally, it may be challenging to consider that intratracheal infusion of bacteria simultaneously with the blocking antibody would be effective. A more detailed experimental plan, including a negative control such as a non-blocking antibody, would be prudent.
8. The demonstration of lung specificity for endothelial CXCR2's effect on neutrophil emigration is not fully convincing without a positive control. An exploration of the underlying mechanism and a discussion on "lung specificity" would be informative.
9. In Figure 6, the identification of groups based on dot size is challenging, and colocalization is difficult to ascertain from the

transmission electron microscopy images. A methodological explanation for dot quantification and whether it was performed blindly by multiple readers would clarify the approach.

10. The study dismisses the role of conventional chemokine signaling in CXCR2-mediated chemokine transcytosis based on a singular experiment. The involvement of Bruton's tyrosine kinase, demonstrated using knockout mice, should be further investigated using additional methods, such as Western blotting, and the participation of other related signaling proteins should be examined.

Reviewer #2

(Remarks to the Author)

In this interesting manuscript, the authors report a novel role for endothelial and epithelial CXCR2 in the chemotactic migration of neutrophils from the bloodstream into the inflamed lung. Specifically, the authors propose that CXCR2 is critical for the transcytosis of chemotactic cytokines across the alveolar barrier. This is a provocative and potentially important concept. To consolidate this hypothesis, the authors should perform additional controls to ensure that what they see is indeed transcellular and not paracellular transport, and to delineate better the nature of this transport process and its regulation by Bruton's tyrosine kinase. Please find my detailed comments below.

Major comments:

1. The proposed role of CXCR2 in chemokine transcytosis across the alveolar barrier is intriguing and provocative. Given its considerable physiological and therapeutic implications, it is adamant to exclude potential confounding factors, specifically the alternative possibility that CXCR2 regulates endothelial/epithelial permeability, thereby mediating paracellular rather than transcellular chemokine transport. This notion is seemingly counterargued by the findings in Suppl. Fig. 4A,B showing no reduction in BALF protein levels in mice with an endothelial- or epithelial-specific loss in CXCR2; yet these findings are notable in seeming disagreement with the earlier report by Reutershan and colleagues (2006) showing that parenchymal CXCR2 expression is required for LPS induced permeability.

The proposed concept would be considerably strengthened by additional data showing the specificity of the translocation from the alveolar to the vascular compartment, e.g. using a streptavidin-coupled protein of similar size as CXCL1 that does NOT translocate into the bloodstream.

2. There are basically two central mechanisms by which transcytosis can occur, caveolar transport or clathrin coated pits. Lung endothelia and alveolar epithelial cells are rich in caveolae, yet immune-EM stainings in the present study point towards a role for clathrin-coated pits. It would be important to elaborate further on the exact organellar route of transcytosis, either by additional EM stainings for caveolar markers, and/or by additional experiments in the transwell bilayer model with respective inhibitors of clathrin-mediated (e.g. by pitstop 2) versus caveolar transcytosis (by cholesterol depletion e.g. with MBCD), as well as dynamin-mediated transcytosis (e.g. by dyngo 4a) in general.

3. Are expression levels of CXCR2 in lung endothelial or epithelial cells differentially regulated in pneumonia?

4. Suppl. Fig. 1: While Suppl Fig. 1B shows convincing evidence for an epithelial-specific loss of CXCR2 in CXCR2f/f/ShhCre/+ mice, the endothelial loss in CXCR2f/f/Cdh5Cre/+ shown in Fig. 1A is less convincing. Specifically, there seems to be little difference between CXCR2 in endothelial cells of CXCR2f/f/Cdh5Cre/+ and CXCR2f/f/Cdh5Cre/- (center panel) and isotype control for CXCR2 is much lower in CXCR2f/f/ShhCre than in the other groups (bottom panel). Was isotype control always done in Cre- mice? This should probably be specified.

5. With respect to Suppl. Fig. 2E the authors suggest ACKR1 staining specifically in venules of the bronchial circulatory tree. However, in mice (in contrast to humans or larger mammals) the bronchial circulation supplies only a miniscule proportion of the pulmonary tissue (the trachea and main bronchi) while the majority of the lung parenchyma is supplied only by the pulmonary circulation (e.g. Mitzner et al., Am J Pathol 2000).

6. The potential regulation of CXCL1 transcytosis by Btk is interesting, but mechanistically underdeveloped in its present form. Which part of the transcytosis process is regulated by Btk and how?

7. Receptor-mediated transcytosis (as suggested here for CXCL1 via CXCR2) requires changes in receptor affinity as a function of the transcytosis process to ensure that the ligand is taken up with high affinity on the one side, but released at a low receptor affinity on the other side of the transcytotic process. While I understand that the delineation of this molecular process is beyond the scope of the present study, it should be discussed as a limitation.

Minor comments:

1. A schematic/graphical abstract of the proposed concept would be helpful

2. Please check the manuscript carefully for language/grammar or missing words (e.g. page 3, line 24: "and in human CXCL8"; page 4, line 5: "changes in vascular permeability and neutrophil (?) suggested"; page 13, line 13: "have also been described not only chemokines"; page 14, line 11 "analogously, is possible"; page 14, line 20: "This associations...")

3. With the possibility to include supplemental data, "data not shown" (e.g. page 5, line 14) has become largely obsolete. As such, please include CXCR2 levels on circulating leukocytes as a suppl. figure, in particular as the preservation of CXCR2 on neutrophils is of critical relevance for the interpretation of the data.

4. Why is protein content in BALF elevated in global CXCR2 ko, but not in endothelial or epithelial specific KO mice? This finding would imply that CXCR2 on e.g. neutrophils is barrier-protective, which seems counterintuitive.

5. Fig. 3F: Genotype should be Cdh5Cre- and Cre+ ("5" is missing)

6. Page 6, line 24: The authors state "Interestingly, the loss of CXCR2 resulted in more dramatic inhibition when neutrophil accumulation in the BALF was assessed suggesting additive roles of vascular and epithelial CXCR2 in CXCL1 induced neutrophil recruitment into alveolar space" – it is not clear to me how the authors come to the conclusion of an additive effect

here based on the different findings by BALF analysis and intravital microscopy. Please clarify.

7. Fig. 4: Title states "Neutrophil recruitment in cremaster and SPLEEN tissue", yet manuscript text and legend state that neutrophil recruitment was assessed in cremaster muscle and kidney.
8. Considerable parts of the discussion circle around the role of ACKR1 in transcytosis; however, as ACKR1 was not really studied (except for some immunohistological images), this aspect should be significantly reduced.
9. Please specify how transmigrating neutrophils were identified and quantified by intravital microscopy? Were individual neutrophils tracked over the entire passage from exiting alveolar capillaries until entry into the alveolar space?
10. The relevance of the proposed concept would be further strengthened by demonstrating its applicability in a second model of pneumonia (*Strep pneumoniae* or IVA) or alternatively a model of sterile lung injury; however, as this is a proof-of-principle study the lack of a second model could also be discussed as a limitation of the present study.

Reviewer #3

(Remarks to the Author)

The manuscript, "Alveolar epithelial and vascular CXCR2 mediates transcytosis of CXCL1 in inflamed lungs" uses comparisons between global CXCR2 vs epithelial CXCR2 (Shh Cre) vs endothelial CXCR2 (Cdh5 Cre) to indicate a novel lung-specific form of chemokine transcytosis dependent on CXCR2 expression in non-hematopoietic cells. They primarily use a model of *Klebsiella* infection in alveolae to investigate CXCL1 distribution and neutrophil recruitment in acute lung inflammation. The paper aims to build upon existing non-canonical functions of chemokines for transcytosis beyond immune cells. This work is interesting and builds upon a small but growing body of evidence that chemokine receptors have non-signaling roles involved in chemoattractant distribution. Publication should be considered, although there are issues that must be addressed:

1. In Figure 5F, it looks like there are two populations when only the endothelium does not have CXCR2, one that lets minimal CXCL1 through and ones more similar to the epithelium knockouts. The text detailing the transwell experiment should address this discrepancy between the CXCR2 KO in both the endothelial and epithelial layer versus only the endothelial layer.
2. In Figure 6C and 6D, it would be helpful to include a cartoon on how the CXCL1 has to first transcytose across the epithelial layer to then reach the endothelial layer. This would make it easier to interpret the data in Figure 6D where CXCL1 accumulation in the endothelial cells is also dependent on epithelial transcytosis first.
3. In Supplementary Figure 1, the flow cytometry data of conditional CXCR2 KO in endothelial cells does not have good/sufficient separation by flow cytometry.
4. In the section where CXCL1 and fMLP are used in intravital microscopy, the function of these chemoattractants should be introduced to underscore why the data is different.
5. In the introduction, this sentence is awkward:

Neutrophil migration into the lungs is tightly controlled involving neutrophils passing the endothelial barrier of alveolar capillaries, a thin interstitial space, and the alveolar epithelial layer to reach the alveolar space.

6. The abbreviation BAL is used without introducing what it is explicitly.

Reviewer #4

(Remarks to the Author)

Reviewer #5

(Remarks to the Author)

The current study adds to previous reports delineating the role of CXCR2 in lung neutrophil recruitment in response to bacterial infection. The topic itself is not novel, and part of the presented data is already published by others. The novel aspect relates to the employment of endothelial and epithelial specific CXCR2 KO mice and its effect on lung neutrophil mobilization in response to *K. pneumoniae*. There are several criticisms related to this work as follows.

Major comments:

- 1) The currently employed Kpn strain ATCC 13883 is relatively avirulent in mice, requiring extremely high CFU counts to trigger respiratory illness. It is most commonly used for antimicrobial testing purposes. In this study, 40 times higher CFU counts as compared to the usually employed 10^6 CFU/mouse of a virulent Kpn strain were applied into the lungs of mice. Such high infection doses however do not reflect the clinical infection setting, while raising questions about the relevance of the reported findings.
- 2) The study design is unclear, as just a 24 h time point post-infection was examined, which does not reflect a typical

infection course in mice. Moreover, CXCR2 KO and corresponding WT mice were infected with just half of the infection dose, ie 2×10^7 CFU/mouse, which makes a direct comparison of data between groups difficult.

3) The employed RB6-8C5 antibody for neutrophil detection via intravital microscopy is not specific for neutrophils, as it also binds to monocyte subsets. The anesthesia and ventilation protocols have not been described.

4) The method of MLMVEC and AEC purifications are missing.

5) Fig. S1: The gating of endothelial cells (lower dot plot) appears to include several populations of CD31 expressing cells. How did the authors exclude CD31 expressing thrombocytes from the gating, and how did they exclude CD31 expressing leukocytes adhering to endothelial cells? Photographic illustrations of the gated endothelial and epithelial cell populations must be shown to validate the presented data. The shift in conditional pulmonary endothelial CXCR2 KOs is not apparent.

6) Legend to Fig. S3: What is a bronchial circulatory tree, and what are alveolar capillaries?

7) The CFU data of Fig. 1/2 are difficult to interpret: Why are CFU counts in global CXCR2 KO mice so much lower as compared to endo/epithelial CXCR2 KOs? The data presentation is in linear scale mode, while just showing CFU per ml, rather than in standard log scale mode depicting whole CFU data per compartment (BAL, lung, blood). The extremely low CFU counts in blood as compared to extremely high CFU counts in lungs are typical of avirulent Kpn strain employed, making the biological relevance questionable. With an estimated 8×10^8 CFU/lung (without BAL) in endothelial CXCR2 KOs and 4×10^8 CFU in controls, it is simply impossible to judge a biologically significant difference in this extremely high CFU range that does not even differ by one order of magnitude.

Fig. 2J-K: What effect did CXCR2 blockade have on bacterial CFU in mice?

8) Fig. 3: From the provided photographic illustrations, it appears that alveolar septa are strongly enlarged in both WT and CXCR2 KO mice challenged with either CXCL1 or fMLP. How does this affect the transit time of PMN crossing the barrier to reach the alveoli, and thus being accessible by BAL? Inclusion of control sections of untreated mice is needed.

9) Did the authors determine CXCL1 levels in plasma of infected WT and global versus endo-/epithelial CXCR2 deficient mice? A global (or partial) chemokine receptor KO should lead to increased levels of the ligand in the circulation upon infection. Against this, why should intratracheal CXCL1 be transcytosed across barrier cells into the circulation despite the fact that just WT mice express CXCR2 on both barrier cell types, which should capture and subsequently internalize the ligand?

10) Given the published cross-reactivity between CC and CXC chemokine receptors in inflammatory monocyte and neutrophil migration to the lung, what effect did the authors note regarding inflammatory monocyte trafficking in the current model?

11) The authors suggest a severe impairment of lung bacterial clearance in selective CXCR2 KO, but this has not been proven in respective survival studies or more detailed infection studies addressing the course of infection over time. This makes the biological significance of the reported findings enigmatic.

Version 1:

Reviewer comments:

Reviewer #1

(Remarks to the Author)

In the second paragraph before introducing ACKR (Atypical Chemokine Receptor 1), the authors should provide a brief description of current knowledge on chemokine transcytosis. This will help to make the narrative more coherent and easier for general readers to understand the study's background. Additionally, they should spell out ACKR1 on its first mention.

Reviewer #2

(Remarks to the Author)

The authors have performed extensive additional experiments to i) demonstrate that CXCL1 transport occurs via the trans- rather than the paracellular route, ii) on the effects of different inhibitors of transcytotic transport in the transwell bilayer model, iii) in bone marrow chimeric Btk-KO mice, and iv) in a second model of E. coli induced bacterial pneumonia. These additional data largely confirm the authors' proposed concept, and have significantly strengthened the paper. A few major and minor concerns, however, persist or arose de novo as a result of the revised graphs and text, and need to be addressed.

Major comments:

1. In response to my previous comment 3 "Are expression levels of CXCR2 in lung endothelial or epithelial cells differentially regulated in pneumonia?" the authors now state that CXCR2 expression was found to be increased in WT mice after pneumonia induction and refer to Supplemental Fig. 1; however, I was unable to detect any flow cytometric data from pneumonia mice in the respective figure.

2. In response to my previous comment 4, the authors replaced the original exemplary flow cytometric data, but unfortunately these new data leave me even more confused: From what I see in Suppl. Fig. 1, it seems that CXCR2 levels decreased in BOTH endothelial and epithelial cells no matter whether the knockout was seemingly specific for endothelial or epithelial cells (notably, the strongest loss of CXCR2 is seen in endothelial cells after knockout of epithelial CXCR2 (panel on the lower left)). This critical aspects needs proper clarification.

3. I am also still confused about the intravital microscopic assessment of "transmigrating neutrophils" (page 7, line 23). As

stated now by the authors “intravital microscopy does not allow the precise allocation of the neutrophils within the different compartments of the lung. Therefore, we analyzed the presence of interacting neutrophils in the lung as cells remaining stationary for >30s in the pulmonary vasculature, the interstitial space or the alveoli.” First, how can you differentiate between stationary cells in the pulmonary vasculature, the interstitial space or the alveoli, when you cannot precisely allocate neutrophils to these different compartments? Second, how does the number of stationary neutrophils translate into counts of “transmigrating neutrophils”? Finally, the authors report that “neutrophils were visualized in the alveoli by intravenous administration of an AF488-coupled Gr-1 antibody before starting microscopy” – how can the antibody reach the neutrophil if the neutrophil is already in the interstitium or the alveolar space?

Minor comments

1. In the new fig. 6G, please indicate in the graph which of the x-axis denominations (+/+, -/-) refer to endothelial and which to epithelial cells.
2. The authors have added new data to explore the regulation of CXCL1 transcytosis by Btk in greater depth, and now show regulation of p38 and Akt phosphorylation by Btk. To make the mechanistic connect, could the authors add a sentence or two how these pathways link to clathrin-coated pits mediated transcytosis?
3. Graphical abstract: In the graphical abstract, it seems as if CXCR2 once it binds CXCL1 is transported across both the epithelium and endothelium. This is likely not the case; rather, CXCR2 will release CXCL1 on the basolateral side of the epithelium where it is then taken up in turn by CXCR2 expressed on the apical side of the endothelium and transcytosed again to be expressed on the vascular surface.
4. Page 14, line 11: Please change “contend” to “content”

Reviewer #3

(Remarks to the Author)

The manuscript, “Alveolar epithelial and vascular CXCR2 mediates transcytosis of CXCL1 in inflamed lungs” uses comparisons between global CXCR2 vs epithelial CXCR2 (Shh Cre) vs endothelial CXCR2 (Cdh5 Cre) to indicate a novel lung-specific form of chemokine transcytosis dependent on CXCR2 expression in non-hematopoietic cells. They primarily use a model of Klebsiella infection in alveolae to investigate CXCL1 distribution and neutrophil recruitment in acute lung inflammation. This work is interesting and builds upon a small but growing body of evidence that chemokine receptors have non-signaling roles involved in chemoattractant distribution. After revisions addressing concerns, publication is recommended after addressing 1 major issue and several minor issues:

1. In Figure 6G, inhibitors for clathrin-, caveolin-, and dynamin-dependent endocytosis/transcytosis are used with the intention to further the transcytosis mechanism used by CXCR2 in the transport of CXCL1. However, the data used for this experiment measures neutrophil recruitment, not CXCL1 distribution. Thus, the data does not directly show that clathrin is necessary for CXCL1 transcytosis rather than neutrophil infiltration. Please directly show CXCL1 changes in distribution using the inhibitors.
2. In Figures 1, 2, and Supplementary Figure 4, please delineate which bars are saline versus infection on the graphs themselves.
3. Address in the discussion an evolution theory/hypothesis as to why CXCR2 on lung endothelial cells and not CXCR2 in the kidney or cremaster are privileged to transcytose CXCL1.

Reviewer #4

(Remarks to the Author)

Reviewer #5

(Remarks to the Author)

My primary concerns have not been addressed:

- 1) The concern of using the anti-Gr-1 antibody clone RB6-8C5 for depletion of neutrophils was related to the previously reported co-depleting activities of this antibody against monocytes. Therefore, it simply makes no sense to perform immunofluorescence analysis of lung tissue collected from mice pre-treated with this antibody while stating that no monocytes were found in these mice at 4 h post-infection. Rather, FACS analyses of lung tissue digests would have been needed to clarify the effect of this antibody on lung monocyte subsets in the given experimental context.
- 2) The biological relevance of the employed model is still not clear. The authors provide survival curves comparing WT with CXCR2 KO mice after challenge with *K. pneumoniae*. However, this experiment was not asked for, as it simply confirms

findings that are already known, i.e. that CXCR2 is relevant for survival of bacterial pneumonia. However, the question was whether selective endo- and epithelial CXCR2 KO would also be relevant to survival of pneumonia? Just these critical survival studies have not been provided by the authors.

3) I was asking for CXCL1 chemokine levels in plasma of CXCR2 KO and partial CXCR2 KO mice (i.e., endo- versus epithelial CXCR2 KO). However, the authors just report plasmatic CXCL1 levels in global CXCR2 KO mice challenged with *K. pneumoniae*. Of note, these data do not add anything novel to the study, nor support the claims of the authors. It is well known that global knockout of a given chemokine receptor increases plasmatic levels of the respective chemokine ligands in infected mice. Considering endothelial CXCR2 KO mice, the concept would be that endothelial CXCR2 deficiency would lead to increased plasmatic CXCL1 levels, which may cause CXCR2 receptor desensitization on the neutrophil surface (which can be determined by FACS analysis) followed by receptor internalization, thereby rendering neutrophils unresponsive to local alveolar CXCL1 chemokine levels. This in turn would promote bacterial pneumonia and ultimately bacterial dissemination and mortality. However, these kind of experiments have not been addressed during revision of the manuscript.

Version 2:

Reviewer comments:

Reviewer #1

(Remarks to the Author)

The manuscript has been significantly improved.

I have one minor correction: please change "makrophage" to "macrophage" in the graphical abstract.

Reviewer #2

(Remarks to the Author)

In response to my previous comments, the authors have performed additional intravital microscopic analyses, corrected the presentation of their flow cytometric analyses, clarified their data presentation for their neutrophil transwell migration assay, and added new text to discuss the role of p38 and Akt in clathrin-coated pits mediated transcytosis. Overall, the authors have adequately addressed all my previous comments.

Reviewer #3

(Remarks to the Author)

The manuscript, "Alveolar epithelial and vascular CXCR2 mediates transcytosis of CXCL1 in inflamed lungs" uses comparisons between global CXCR2 vs epithelial CXCR2 (Shh Cre) vs endothelial CXCR2 (Cdh5 Cre) to indicate a novel lung-specific form of chemokine transcytosis dependent on CXCR2 expression in non-hematopoietic cells. They primarily use a model of *Klebsiella* infection in alveolae to investigate CXCL1 distribution and neutrophil recruitment in acute lung inflammation. This work is interesting and builds upon a small but growing body of evidence that chemokine receptors have non-signaling roles involved in chemoattractant distribution. After revisions addressing all concerns, publication is recommended.

Reviewer #4

(Remarks to the Author)

Reviewer #1

The study under review investigates the role of CXCR2 in neutrophil recruitment and bacterial clearance in the lung, utilizing global and cell-specific conditional KO mice. The findings suggest that CXCR2 deficiency hampers neutrophil recruitment and increases bacterial burden in all KO strains, with CXCR2 neutralization reducing neutrophil recruitment. The authors also report that pulmonary endothelial and epithelial CXCR2 support neutrophil recruitment, and KO mice exhibit decreased neutrophil transmigration upon CXCL1 stimulation. Using in vitro studies with a Transwell™ model, reduced neutrophil transmigration through endothelial-epithelial bilayers lacking CXCR2 is demonstrated. Additionally, transmission electron microscopy reveals CXCL1 clustering with CXCR2 in lung epithelial and endothelial cells during infection. In vivo experiments further show decreased CXCL1 transcytosis in CXCR2-deficient mice, implicating the involvement of Bruton's tyrosine kinase. The authors conclude that CXCR2 plays a role in alveolar epithelial and endothelial cells by mediating cognate CXCL1 transcytosis, favoring neutrophil recruitment to infected lungs. While the study addresses an important topic, it predominantly relies on knockout mouse models, and there are notable gaps in experimental controls. The rationale behind the experimental design and the mechanistic pathway proposed require further elucidation.

Response: We thank the reviewer for the thoughtful evaluation of our work.

Below are detailed comments for consideration:

1. The manuscript would benefit from a more comprehensive characterization of the global and cell-specific CXCR2 knockout mice. The absence of protein expression data, relative to wild-type mice, and the lack of references for validation are notable omissions. Given the potential for compensatory genetic responses, additional validation experiments using blocking antibodies (or siRNA) against CXCR2 are recommended to substantiate the findings in these knockout mice.

Response: We agree with the reviewer and performed additional experiments to further characterize the global and conditional CXCR2 knockout mouse strains.

The following paragraph was added to the results section (page 5, line 17):

“In addition, the surface expression of various cell surface markers and the intracellular expression of molecules involved in endocytosis and transcytosis pathways did not differ on pulmonary endothelial and epithelial cells isolated from the experimental mouse strains (Supplemental Figure 4A-C and Figure 7B).”

The following paragraph was added to the results section (page 6, line 1):

“Intratracheal instillation of *K. pneumoniae* resulted in recruitment of neutrophils into the lungs of WT control mice compared to saline-treated control mice and was significantly reduced in all CXCR2 knockout strains tested and WT that received a blocking CXCR2 antibody intratracheally (Figure 1A, D, G and Supplemental Figure 5A-H). The baseline and post-infection experiments were conducted concurrently. Impaired neutrophil recruitment in knockout mice and after intratracheal instillation of a blocking CXCR2 antibody was observed in the whole lung tissue and also in the bronchoalveolar lavage (BAL, Figure 2A, D, G). In all knockout mouse strains and WT mice who received a blocking CXCR2 antibody intratracheally, a decrease in neutrophil recruitment was associated with a significant increase in bacterial burden, as e.g. determined by the quantification of bacterial colony forming units (CFUs), in the lung and the spleen (Figure 1B, C, E, F, H, I and Supplemental Figure 5A-H). This effect was further evident in the BAL and the blood (Figure 2B, C, E, F, H, I).”

The following paragraph was added to the results section (page 11, line 3):

“Of note, the expression of the endocytosis/transcytosis-related molecules clathrin, caveolin and dynamin was not significantly different among the different knockout mouse strains (Figure 7B).”

2. The assertion that conditional CXCR2 knockout in endothelial and epithelial cells does not impact CXCR2 expression in circulating leukocytes warrants supporting evidence. The data should be included as supplementary material.

Response: We agree with the reviewer and quantified CXCR2 expression on circulating leukocytes in WT, global CXCR2 knockout as well as in conditional CXCR2 knockout mice by flow cytometry.

The following paragraph was added to the results section (page 5, line 14):

“Conditional CXCR2 knockout in endothelial and epithelial cells did not affect CXCR2 expression on circulating leukocytes (Supplemental Figure 2).”

3. The study assesses ACKR1 expression solely via immunostaining, without a clear rationale for its relevance to the study's objectives. Further clarification on the reasoning behind this measurement and additional data to confirm the absence of this protein is necessary.

Response: We thank the reviewer for addressing this point.

We included the following paragraph in the introduction to further elaborate the rationale for analyzing ACKR1 expression in the lung (page 3, line 22):

“ACKR1 has been shown to decisively contribute to the endothelial cell binding, internalization and transcytosis of chemokines, thus supporting their function in leukocyte extravasation (Middleton 2005, Pruenster 2009). However, within the vascular tree ACKR1 is normally expressed only in the postcapillary and collective venules and small veins and is conspicuously missing in the capillaries throughout all organs and tissues (Thiriot 2017), including alveolar capillaries (Lee 2003, Peiper 1995). Therefore, ACKR1 cannot contribute to the chemokine transcytosis and immobilization in the alveolar vascular network and thus support chemokine driven leukocyte emigration into the alveolar spaces. Thus, we set to investigate if an alternative chemokine transcytosis pathway exists in the alveolar vessels.”

We included the following paragraph in the discussion (page 15, line 2):

“We show ACKR1 protein expression by immunohistochemistry using a fully validated highly specific antibody. This is the best and only way to analyze the very sparse ACKR1 expression in the lung as even scRNA approaches miss ACKR1 signals because the population of ACKR1-positive cells is so rare.”

4. The lung infection model lacks critical negative controls, such as wild-type, Cre-positive, and Cre-negative control mice treated with phosphate-buffered saline, the solvent for intratracheal bacterial infusion. Conducting statistical analysis across all four groups post-experiment and replicating the experiments with blocking antibodies would strengthen the study's validity.

Response: We agree with the reviewer and performed additional experiments with saline-treated control animals as well as animals treated with blocking anti-CXCR2 antibodies. Statistical analysis was performed on all groups (including saline-treated control mice as well as anti-CXCR2 antibody-treated mice) post-experiment by one-way ANOVA analysis and post-hoc testing. Statistical significance is indicated in the respective figures.

The following paragraph was added to the results section (page 6, line 1):

“Intratracheal instillation of *K. pneumoniae* resulted in recruitment of neutrophils into the lungs of WT control mice compared to saline-treated control mice and was significantly reduced in all CXCR2 knockout strains tested and WT that received a blocking CXCR2 antibody intratracheally (Figure 1A, D, G and Supplemental Figure 5A-H). The baseline and post-infection experiments were conducted concurrently. Impaired neutrophil recruitment in knockout mice and after intratracheal instillation of a blocking CXCR2 antibody was observed in the whole lung tissue and also in the bronchoalveolar lavage (BAL, Figure 2A, D, G). In all knockout mouse strains and WT mice who received a blocking CXCR2 antibody intratracheally, a decrease in neutrophil recruitment was associated with a significant increase in bacterial burden, as e.g. determined by the quantification of bacterial colony forming unites (CFUs), in the lung and the spleen (Figure 1B, C, E, F, H, I and Supplemental Figure 5A-H). This effect was further evident in the BAL and the blood (Figure 2B, C, E, F, H, I).”

5. Discrepancies between Figure 1, which depicts two mouse groups treated with K. pneumoniae, and supplementary Figure 4, which includes four groups, raise questions. Clarification on whether the baseline and post-infection experiments were conducted concurrently would be beneficial.

Response: We agree with the reviewer and now also included data from saline-treated control mice in Figure 1 and 2 (see also comments above). The baseline and post-infection experiments were conducted concurrently.

The following paragraph was added to the results section (page 6, line 5):

“The baseline and post-infection experiments were conducted concurrently.”

6. *Inconsistencies in the number of mice evaluated within the same group, as evidenced by varying data points across figures, necessitate an explanation or justification for any data exclusion. For example, in the CXCR2^{fl/fl}Cdh5Cre⁺ group, the number of dot plots varies between Fig 1D (n=7), Fig 1E (n=9), and Fig 1F (n=13).*

Response: We agree with the reviewer that the numbers of mice in the different groups are inconsistent. Therefore, we did additional experiments to include the same number of mice in each group.

7. *The administration route for the CXCR2 blocking antibody, used concurrently with bacterial infusion, requires clarification. If administered intratracheally, it may be challenging to consider that intratracheal infusion of bacteria simultaneously with the blocking antibody would be effective. A more detailed experimental plan, including a negative control such as a non-blocking antibody, would be prudent.*

Response: We agree with the reviewer and included additional experiments with non-blocking control antibodies to exclude any unwanted effects of the co-administration of the antibodies together with the bacteria.

The following paragraph was added to the results section (page 6, line 13):

“The concurrent intratracheal administration of a non-blocking control antibody did not show any significant effects on neutrophil recruitment and CFU counts (Supplemental Figure 5A-H).”

8. *The demonstration of lung specificity for endothelial CXCR2's effect on neutrophil emigration is not fully convincing without a positive control. An exploration of the underlying mechanism and a discussion on "lung specificity" would be informative.*

Response: We agree with the reviewer and performed additional experiments.

The following paragraph was added to the results section (page 8, line 8):

“To assess if the impact of endothelial CXCR2 on neutrophil emigration is lung specific, neutrophil recruitment was analyzed in the murine cremaster muscle of endo-

thelial specific CXCR2 knockout mice (CXCR2^{fl/fl}Cdh5^{Cre+}) and appropriate WT controls by intravital microscopy. For this purpose, mice were challenged with either an intrascrotal injection of TNF α or with a local superfusion of CXCL1 (for 2h) or fMLP (for 5 min) before analyzing neutrophil adherence and emigration. The ablation of endothelial CXCR2 did not affect neutrophil adherence or emigration in the cremaster muscle following TNF α -stimulation, fMLP or CXCL1 superfusion (Figure 4A-F).”

9. In Figure 6, the identification of groups based on dot size is challenging, and colocalization is difficult to ascertain from the transmission electron microscopy images. A methodological explanation for dot quantification and whether it was performed blindly by multiple readers would clarify the approach.

Response: We agree with the reviewer and revised the description in the methods section. Furthermore, we added a cartoon depicting the relevant structures in Figure 6.

The following paragraph was added to the methods section (page 23, line 20):
“Single electron microscopic images were analyzed for distribution of CXCL1 (12nm sized gold particles) on endothelial and epithelial cells in a double-blinded manner by multiple researchers.”

10. The study dismisses the role of conventional chemokine signaling in CXCR2-mediated chemokine transcytosis based on a singular experiment. The involvement of Bruton's tyrosine kinase, demonstrated using knockout mice, should be further investigated using additional methods, such as Western blotting, and the participation of other related signaling proteins should be examined.

Response: We agree with the reviewer and performed additional experiments.

The following paragraph was added to the results section (page 11, line 21):
“On a molecular level, the loss of Btk led to reduced phosphorylation of p38 and Akt in isolated alveolar epithelial cells after stimulation with KC (Figure 7C). To demonstrate the functional relevance of BTK *in vivo*, WT and Btk-KO recipient mice were lethally irradiated for total ablation of the native bone marrow and consecutively reconstituted with isolated bone marrow cells from WT donor mice. After induction of

bacterial pneumonia, the loss of non-hematopoietic Btk expression led to defective neutrophil recruitment into the lung and increased CFU counts (Figure 7D-G).

Reviewer #2

In this interesting manuscript, the authors report a novel role for endothelial and epithelial CXCR2 in the chemotactic migration of neutrophils from the bloodstream into the inflamed lung. Specifically, the authors propose that CXCR2 is critical for the transcytosis of chemotactic cytokines across the alveolar barrier. This is a provocative and potentially important concept. To consolidate this hypothesis, the authors should perform additional controls to ensure that what they see is indeed transcellular and not paracellular transport, and to delineate better the nature of this transport process and its regulation by Bruton's tyrosine kinase. Please find my detailed comments below.

Response: We thank the reviewer for the thoughtful evaluation of our work.

Major comments:

1. The proposed role of CXCR2 in chemokine transcytosis across the alveolar barrier is intriguing and provocative. Given its considerable physiological and therapeutic implications, it is adamant to exclude potential confounding factors, specifically the alternative possibility that CXCR2 regulates endothelial/epithelial permeability, thereby mediating paracellular rather than transcellular chemokine transport. This notion is seemingly counterargued by the findings in Suppl. Fig. 4A,B showing no reduction in BALF protein levels in mice with an endothelial- or epithelial-specific loss in CXCR2; yet these findings are notable in seeming disagreement with the earlier report by Reutershan and colleagues (2006) showing that parenchymal CXCR2 expression is required for LPS induced permeability. The proposed concept would be considerably strengthened by additional data showing the specificity of the translocation from the alveolar to the vascular compartment, e.g. using a streptavidin-coupled protein of similar size as CXCL1 that does NOT translocate into the bloodstream.

Response: We thank the reviewer for the thoughtful comment and performed exactly the experiment that the reviewer suggested.

The following paragraph was added to the results section (page 6, line 24):

“Likewise, the intratracheal co-administration of CXCL1 (MW 8 kD) together with FITC-labeled Dextran (MW 10 kD) showed no significant extravasation of FITC-Dextran into the circulation *in vivo*, thus excluding the possibility of CXCR2 transport by increased vascular permeability rather than transcytosis (Supplemental Figure 7D-E).”

2. There are basically two central mechanisms by which transcytosis can occur, caveolar transport or clathrin coated pits. Lung endothelia and alveolar epithelial cells are rich in caveolae, yet immune-EM stainings in the present study point towards a role for clathrin-coated pits. It would be important to elaborate further on the exact organellar route of transcytosis, either by additional EM stainings for caveolar markers, and/or by additional experiments in the transwell bilayer model with respective inhibitors of clathrin-mediated (e.g. by pitstop 2) versus caveolar transcytosis (by cholesterol depletion e.g. with MBCD), as well as dynamin-mediated transcytosis (e.g. by dyngo 4a) in general.

Response: We thank the reviewer for the valuable suggestions and performed additional experiments in the transwell bilayer model with the suggested inhibitors pitstop 2, MBCD and dyngo 4a.

The following paragraph was added to the results section (page 10, line 14):

“To further elaborate on the exact organellar route of transcytosis, we performed transmigration assays using the transwell bilayer model with inhibitors of clathrin- (pitstop 2), caveolin- (MBCD) or dynamin-mediated transcytosis (dyngo 4a) and observed a significant reduction in neutrophil transmigration after inhibition of clathrin-mediated transcytosis (Figure 6G).”

3. Are expression levels of CXCR2 in lung endothelial or epithelial cells differentially regulated in pneumonia?

Response: We thank the reviewer for pointing out this interesting aspect and performed additional experiments to investigate the expression levels of CXCR2 on lung endothelial and epithelial cells after induction of bacterial pneumonia.

The following paragraph was added to the results section (page 5, line 9):

“The absence/expression of CXCR2 from all lung cells in CXCR2^{-/-} (global knockout) mice as well as cell type-specific absence in lung endothelial cells in CXCR2^{fl/fl}Cdh5^{Cre+} and epithelial cells in CXCR2^{fl/fl}Shh^{Cre+} mice was validated by flow cytometry following IL-1 β stimulation and found to be increased in WT mice after pneumonia induction (Supplemental Figure 1).”

4. Suppl. Fig. 1: While Suppl Fig. 1B shows convincing evidence for an epithelial-specific loss of CXCR2 in CXCR2^{fl/fl}/ShhCre/+ mice, the endothelial loss in CXCR2^{fl/fl}/Cdh5Cre/+ shown in Fig. 1A is less convincing. Specifically, there seems to be little difference between CXCR2 in endothelial cells of CXCR2^{fl/fl}/Cdh5Cre/+ and CXCR2^{fl/fl}/Cdh5Cre/- (center panel) and isotype control for CXCR2 is much lower in CXCR2^{fl/fl}/ShhCre than in the other groups (bottom panel). Was isotype control always done in Cre- mice? This should probably be specified.

Response: We agree with the reviewer and show better exemplary flow cytometry images in Supplemental Figure 1 and specified the respective control mice more precisely.

The following paragraph was added to the results section (page 5, line 13):

“Isotype controls in conditional mouse strains were always performed in Cre-negative mice.”

5. With respect to Suppl. Fig. 2E the authors suggest ACKR1 staining specifically in venules of the bronchial circulatory tree. However, in mice (in contrast to humans or larger mammals) the bronchial circulation supplies only a miniscule proportion of the pulmonary tissue (the trachea and main bronchi) while the majority of the lung parenchyma is supplied only by the pulmonary circulation (e.g. Mitzner et al., Am J Pathol 2000).

Response: We thank the reviewer for pointing out this interesting aspect.

The following paragraph was added to the discussion (page 15, line 2):

“We show ACKR1 protein expression by immunohistochemistry using a fully validated highly specific antibody. This is the best and only way to analyze the very sparse

ACKR1 expression in the lung as even scRNA approaches miss ACKR1 signals because the population of ACKR1-positive cells is so rare. Furthermore, in mice (in contrast to humans or larger mammals) the bronchial circulation supplies only a miniscule proportion of the pulmonary tissue (the trachea and main bronchi) while the majority of the lung parenchyma is supplied only by the pulmonary circulation (Mitzner 2000)."

6. The potential regulation of CXCL1 transcytosis by Btk is interesting, but mechanistically underdeveloped in its present form. Which part of the transcytosis process is regulated by Btk and how?

Response: We agree with the reviewer and performed additional experiments to further investigate the regulation of transcytosis by Btk and the pathophysiological relevance of Btk in the pneumonia model *in vivo*.

The following paragraph was added to the results section (page 11, line 23):

"To demonstrate the functional relevance of BTK *in vivo*, WT and Btk-KO recipient mice were lethally irradiated for total ablation of the native bone marrow and consecutively reconstituted with isolated bone marrow cells from WT donor mice. After induction of bacterial pneumonia, the loss of non-hematopoietic Btk expression led to defective neutrophil recruitment into the lung and increased CFU counts (Figure 7D-G)."

The following paragraph was added to the results section (page 11, line 21):

"On a molecular level, the loss of Btk led to reduced phosphorylation of p38 and Akt in isolated alveolar epithelial cells after stimulation with KC (Figure 7C)."

7. Receptor-mediated transcytosis (as suggested here for CXCL1 via CXCR2) requires changes in receptor affinity as a function of the transcytosis process to ensure that the ligand is taken up with high affinity on the one side but released at of a low receptor affinity on the other side of the transcytotic process. While I understand that the delineation of this molecular process is beyond the scope of the present study, it should be discussed as a limitation.

Response: We thank the reviewer for emphasizing this interesting aspect.

The following paragraph was added to the discussion (page 17, line 15):

“Receptor-mediated transcytosis (as suggested here for CXCL1 via CXCR2) requires changes in receptor affinity as a function of the transcytosis process to ensure that the ligand is taken up with high affinity on the one side but released with a lower receptor affinity on the other side of the lung interstitium. However, the investigation of the molecular mechanisms guiding these processes is beyond the scope of this study and warrants further research.”

Minor comments:

1. A schematic/graphical abstract of the proposed concept would be helpful

Response: We agree with the reviewer and now provide a graphical abstract

2. Please check the manuscript carefully for language/grammar or missing words (e.g. page 3, line 24: “and in humanS CXCL8”; page 4, line 5: “changes in vascular permeability and neutrophil (?) suggested”; page 13, line 13: “have also been described not only chemokines”; page 14, line 11 “analogously, is possible”; page 14, line 20: “This associations...”

Response: We thank the reviewer for the careful reading of our manuscript and have thoroughly revised the text with respect to correct language and grammar.

3. With the possibility to include supplemental data, “data not shown” (e.g. page 5, line 14) has become largely obsolete. As such, please include CXCR2 levels on circulating leukocytes as a suppl. figure, in particular as the preservation of CXCR2 on neutrophils is of critical relevance for the interpretation of the data.

Response: We agree with the reviewer and included the respective data in the manuscript.

The following paragraph was added to the results section (page 5, line 14):

“Conditional CXCR2 knockout in endothelial and epithelial cells did not affect CXCR2 expression on circulating leukocytes (Supplemental Figure 2).”

4. *Why is protein content in BALF elevated in global CXCR2 ko, but not in endothelial or epithelial specific KO mice? This finding would imply that CXCR2 on e.g. neutrophils is barrier-protective, which seems counterintuitive.*

Response: We agree with the reviewer and added this aspect to the discussion.

The following paragraph was added to the discussion (page 14, line 10):

“Interestingly, we found that the BALF protein content is elevated in global CXCR2^{-/-} mice, but not in conditional endothelial or epithelial KO mice. This finding might be explained by the fact, that additional CXCR2⁺ cells, e.g. tissue resident mast or dendritic cells, are involved in the regulation of pulmonary permeability but not in the process of CXCR2-mediated chemokine transcytosis.”

5. *Fig. 3F: Genotype should be Cdh5Cre⁻ and Cre⁺ (“5” is missing)*

Response: We thank the reviewer for the careful reading of the manuscript and corrected the mistake accordingly.

6. *Page 6, line 24: The authors state “Interestingly, the loss of CXCR2 resulted in more dramatic inhibition when neutrophil accumulation in the BALF was assessed suggesting additive roles of vascular and epithelial CXCR2 in CXCL1 induced neutrophil recruitment into alveolar space” – it is not clear to me how the authors come to the conclusion of an additive effect here based on the different findings by BALF analysis and intravital microscopy. Please clarify.*

Response: We agree with the reviewer and removed the respective statement in the revised version of the manuscript.

7. *Fig. 4: Title states “Neutrophil recruitment in cremaster and SPLEEN tissue”, yet manuscript text and legend state that neutrophil recruitment was assessed in cremaster muscle and kidney.*

Response: We apologize for the mistake. Indeed, we analyzed neutrophil recruitment in the cremaster muscle and the kidney and not in the spleen. We corrected the mistakes in the figure legend and in the figure file accordingly.

8. Considerable parts of the discussion circle around the role of ACKR1 in transcytosis; however, as ACKR1 was not really studied (except for some immunohistological images), this aspect should be significantly reduced.

Response: We agree with the reviewer and shortened the discussion of the role of ACKR1 in transcytosis in the revised manuscript as suggested.

9. Please specify how transmigrating neutrophils were identified and quantified by intravital microscopy? Were individual neutrophils tracked over the entire passage from exiting alveolar capillaries until entry into the alveolar space?

Response: We thank the reviewer for highlighting this aspect and added the requested information to the method description.

The following paragraph was added to the methods section (page 20, line 24):
“The intravital microscopy by fluorescence microscopy does not allow the precise allocation of the neutrophils within the different compartments of the lung. Therefore, we analyzed the presence of interacting neutrophils in the lung as cells remaining stationary for >30s in the pulmonary vasculature, the interstitial space or the alveoli. The number of cells was counted per field of view (FOV).”

10. The relevance of the proposed concept would be further strengthened by demonstrating its applicability in a second model of pneumonia (Strep pneumoniae or IVA) or alternatively a model of sterile lung injury; however, as this is a proof-of-principle study the lack of a second model could also be discussed as a limitation of the present study.

Response: We agree with the reviewer and added a second model of bacterial pneumonia by intratracheal instillation of viable *E. coli* to the study (as also requested by reviewer 5).

The following paragraph was added to the results section (page 6, line 16):
“As a proof-of-concept, we additionally induced bacterial pneumonia by intratracheal instillation of *E. coli* (ATCC 25922) instead of *K. pneumoniae* and observed similar results (Supplemental Figure 6).”

Reviewer #3

The manuscript, “Alveolar epithelial and vascular CXCR2 mediates transcytosis of CXCL1 in inflamed lungs” uses comparisons between global CXCR2 vs epithelial CXCR2 (Shh Cre) vs endothelial CXCR2 (Cdh5 Cre) to indicate a novel lung-specific form of chemokine transcytosis dependent on CXCR2 expression in non-hematopoietic cells. They primarily use a model of Klebsiella infection in alveolae to investigate CXCL1 distribution and neutrophil recruitment in acute lung inflammation. The paper aims to build upon existing non-canonical functions of chemokines for transcytosis beyond immune cells. This work is interesting and builds upon a small but growing body of evidence that chemokine receptors have non-signaling roles involved in chemoattractant distribution. Publication should be considered, although there are issues that must be addressed:

Response: We thank the reviewer for the thoughtful evaluation of our work.

1. In Figure 5F, it looks like there are two populations when only the endothelium does not have CXCR2, one that lets minimal CXCL1 through and ones more similar to the epithelium knockouts. The text detailing the transwell experiment should address this discrepancy between the CXCR2 KO in both the endothelial and epithelial layer versus only the endothelial layer.

Response: We thank the reviewer for pointing out this aspect.

The following paragraph was added to the results section (page 9, line 16):

“Of note, two populations appeared to be present in the group with WT epithelium and CXCR2^{-/-} endothelium. However, this discrepancy was neither statistically significant within the group nor when tested against the other groups.”

2. In Figure 6C and 6D, it would be helpful to include a cartoon on how the CXCL1 has to first transcytose across the epithelial layer to then reach the endothelial layer. This would make it easier to interpret the data in Figure 6D where CXCL1 accumulation in the endothelial cells is also dependent on epithelial transcytosis first.

Response: We agree with the reviewer and added a cartoon illustrating the experimental setup (Figure 6D).

3. In Supplementary Figure 1, the flow cytometry data of conditional CXCR2 KO in endothelial cells does not have good/sufficient separation by flow cytometry.

Response: We agree with the reviewer and show better exemplary flow cytometry images in Supplemental Figure 1.

4. In the section where CXCL1 and fMLP are used in intravital microscopy, the function of these chemoattractants should be introduced to underscore why the data is different.

Response: We agree with the reviewer expanded the description in the results section.

The following paragraph was added to the results section (page 7, line 19):

“Whereas CXCL1 is a CXCR2 ligand, the chemotactic peptide fMLP does not bind to CXCR2 but acts via binding to the receptors FPR1 and FPR2.”

5. In the introduction, this sentence is awkward:

Neutrophil migration into the lungs is tightly controlled involving neutrophils passing the endothelial barrier of alveolar capillaries, a thin interstitial space, and the alveolar epithelial layer to reach the alveolar space.

Response: We agree with the reviewer and rephrased the sentence.

The following paragraph was added to the introduction (page 3, line 9):

“Neutrophil migration into the lungs is a tightly controlled process of subsequent events, involving neutrophils passing the endothelial barrier of alveolar capillaries, a thin interstitial space, and afterwards the alveolar epithelial layer to reach the alveolar space.”

6. *The abbreviation BAL is used without introducing what it is explicitly.*

Response: We thank the reviewer for the careful reading and now introduce the abbreviation BAL (bronchoalveolar lavage) at first mentioning.

The following paragraph was added to the results section (page 6, line 6):

“Impaired neutrophil recruitment in knockout mice was observed in the whole lung tissue and also in the bronchoalveolar lavage (BAL, Figure 2A, D, G).”

Reviewer #4

Response: We thank the reviewer for the thoughtful evaluation of our work.

Reviewer #5

The current study adds to previous reports delineating the role of CXCR2 in lung neutrophil recruitment in response to bacterial infection. The topic itself is not novel, and part of the presented data is already published by others. The novel aspect relates to the employment of endothelial and epithelial specific CXCR2 KO mice and its effect on lung neutrophil mobilization in response to K. pneumoniae. There are several criticisms related to this work as follows.

Response: We thank the reviewer for the thoughtful evaluation of our work.

Major comments:

1) The currently employed Kpn strain ATCC 13883 is relatively avirulent in mice, requiring extremely high CFU counts to trigger respiratory illness. It is most commonly used for antimicrobial testing purposes. In this study, 40 times higher CFU counts as compared to the usually employed 10^6 CFU/mouse of a virulent Kpn strain were applied into the lungs of mice. Such high infection doses however do not reflect the clinical infection setting, while raising questions about the relevance of the reported findings.

Response: The reviewer addressed a valid point, but we (Rossaint, J Exp Med, 2021;218(7):e20201353) and other groups (Lee, FEMS Microbiol Lett 2012;331(1):17-24) previously used this Kpn strain in different studies to investigate the molecular mechanisms of neutrophil recruitment and host immune defense. In addition, we performed additional proof-of-concept experiments using a model of pulmonary infection by intratracheal injection with the *E. coli* strain ATCC 25922 to demonstrate the pathophysiological relevance for an additional clinically represented and relevant bacterial strain. These experiments revealed similar results compared to the Kpn infection model, thus confirming the molecular mechanism of CXCR2-mediated transcytosis in the lung.

The following paragraph was added to the results section (page 6, line 16):

“As a proof-of-concept, we additionally induced bacterial pneumonia by intratracheal instillation of *E. coli* (ATCC 25922) instead of *K. pneumoniae* and observed similar results (Supplemental Figure 6).”

2) *The study design is unclear, as just a 24 h time point post-infection was examined, which does not reflect a typical infection course in mice. Moreover, CXCR2 KO and corresponding WT mice were infected with just half of the infection dose, ie 2×10^7 CFU/mouse, which makes a direct comparison of data between groups difficult.*

Response: We agree with the reviewer and harmonized the infection dose in the revised manuscript.

The following paragraph was added to the discussion (page 17, line 21):

“As a limitation, we have only analyzed neutrophil recruitment and bacterial CFU counts 24 hours after infection. This time point is commonly used in studies focusing on the investigation of specific molecular mechanisms involved in the regulation of leukocyte activation and recruitment (Rossaint, J Exp Med 2021; Conrad, JCI Insight 2022, Ittner, J Exp Med 2012) and the infection course and bacterial invasion behavior of the bacteria itself was not the focus of this study.”

3) *The employed RB6-8C5 antibody for neutrophil detection via intravital microscopy is not specific for neutrophils, as it also binds to monocyte subsets. The anesthesia and ventilation protocols have not been described.*

Response: We agree with the reviewer and performed additional experiments to show that monocytes are not present in relevant numbers in the lung at the relevant time points. Furthermore, we included additional information on the anesthesia and ventilation protocols in the methods section.

The following paragraph was added to the results section (page 8, line 2):

“To exclude that the used Gr1 antibody (clone RB6-8C5) leads to co-labeling of monocytes, we performed additional lung intravital microscopy experiments with an Alexa568-labeled anti-CD115 antibody specifically labeling monocytes after CXCL1 instillation and could not detect significant numbers of monocytes in the lung after 4 hours (Supplemental Figure 9B).”

The following paragraph was added to the methods section (page 20, line 12):

“Briefly, mice were anaesthetized by intraperitoneal injection of ketamine (125 μ g/g body weight, WDT) and xylazine (12.5 μ g/g body weight; Bayer). An endotracheal

tube was placed directly into the trachea via a tracheotomy and the mice were ventilated mechanically with a MiniVent Type 845 small rodent ventilator (Harvard Apparatus, tidal volume 10 μ l/g bodyweight, respiratory frequency 150/min, PEEP 5cmH₂O).”

4) *The method of MLMVEC and AEC purifications are missing.*

Response: We thank the reviewer for the comment and added the method describing the MLMVEC and AEC isolation process to the method section.

The following paragraph was added to the methods section (page 21, line 4):

“Isolation of murine lung microvascular endothelial cells (MLMVECs). Mice were sacrificed by cervical dislocation, the lung lobes were removed and mechanically minced. The tissue homogenisates was enzymatically digested with collagenase A (Roche) and DNase 1 (Merck) at 37°C for 90 minutes with gentle agitation before passing the cell suspension over a 70 μ m cell strainer and centrifugation at 300xg for 5 minutes. 20 μ l magnetic dynabeads (Invitrogen) were incubated with 15 μ g anti-CD31 antibody (clone Mec13.3, BD Biosciences) overnight at 4°C on a shaker. After washing, the dynabeads were incubated with the cell suspension for 45 minutes at 4°C on a shaker. The dynabead-bound cells were passed six times over a magnetic separator (Milltenyi), resuspended in DMEM medium (supplemented with 20% FCS) and seeded on gelatin-coated cell culture dishes. Cells were kept in an cell culture incubator at 37 °C and 10% CO₂. Dynabead-assisted cell sorting for purification was repeated twice per week for a total of 4-5 purification cycles.”

The following paragraph was added to the methods section (page 21, line 18):

“Isolation of alveolar epithelial cells (AECs). After sacrifice the thorax was opened and the trachea was intubated with a 20G plastic canula. The lung was filled with 1ml Dispase solution, followed by 0.5ml of a 1% low-melt agarose (Biozym) heated to 45°C. After 2 minutes of external cooling of the lung, the lung lobes are removed, placed into 0.5ml of Dispase and incubated at 25°C for 45 min with gentle agitation. Afterwards, the lung was minced into small pieces and serially passed over cell strainers with 40 and 20 μ m pore size. The cell suspension was centrifuged at 200xg for 7 min and incubated in erythrocyte lysis buffer supplemented with 10 μ l DNase I (Merck) for 2 min before centrifugation at 300xg für 5 min. The pellet was resuspended in DMEM supplemented with 10% FCS, 1% Penicillin/Streptomycin and 1% Glutamine.

Subsequently, negative selection was performed with 1 μ g of biotinylated CD45 antibody (clone 30-F11, BD Biosciences) and 0.65 μ g of biotinylated CD16/32 antibody (clone 2.4G2, BD Biosciences) per 1x10⁶ cells and incubated with 100 μ l washed MagneSpheres (Promega) for 30 min. The cell suspension was passed three times over a magnetic separator for 3 min and the supernatant was collected. The cells were seeded on cell culture dishes coated with 15 μ g/ml fibronectin (Merck) and incubated at 37°C and 5% CO₂.”

5) Fig. S1: The gating of endothelial cells (lower dot plot) appears to include several populations of CD31 expressing cells. How did the authors exclude CD31 expressing thrombocytes from the gating, and how did they exclude CD31 expressing leukocytes adhering to endothelial cells? Photographic illustrations of the gated endothelial and epithelial cell populations must be shown to validate the presented data. The shift in conditional pulmonary endothelial CXCR2 KOs is not apparent.

Response: We thank the reviewer for pointing out these important aspects. Platelets were excluded by prior gating out small (PLT-sized) events in the native SSC/FSC scatter plot. CD31 expressing leukocytes were excluded by gating on the CD45 negative population.

We included additional examples of the gating strategy in the revised Supplemental Figure 1.

6) Legend to Fig. S3: What is a bronchial circulatory tree, and what are alveolar capillaries?

Response: We thank the reviewer for pointing out these misleading expressions and changed the description in the manuscript accordingly.

The following paragraph was added to the discussion (page 15, line 9):

“However, in chronic lung inflammatory diseases ACKR1 expression is induced in broader segments of vessels lining the pulmonary bronchiolar tree, possibly also in vessel directly lining the alveolae.”

The following paragraph was added to the figure legends:

“ACKR1 staining (green) in venules lining the bronchial tree in the submucosa of a large bronchus (bronchial lumen outlined by dotted line).”

7) The CFU data of Fig. 1/2 are difficult to interpret: Why are CFU counts in global CXCR2 KO mice so much lower as compared to endo/epithelial CXCR2 KOs? The data presentation is in linear scale mode, while just showing CFU per ml, rather than in standard log scale mode depicting whole CFU data per compartment (BAL, lung, blood). The extremely low CFU counts in blood as compared to extremely high CFU counts in lungs are typical of avirulent Kpn strain employed, making the biological relevance questionable. With an estimated 8×10^8 CFU/lung (without BAL) in endothelial CXCR2 KOs and 4×10^8 CFU in controls, it is simply impossible to judge a biologically significant difference in this extremely high CFU range that does not even differ by one order of magnitude. Fig. 2J-K: What effect did CXCR2 blockade have on bacterial CFU in mice?

Response:

We thank the reviewer for pointing out these important aspects. We have chosen an established bacterial pneumonia model using intratracheal instillation of the ATCC 13883 to study the molecular mechanisms of chemokine transcytosis and the regulation of neutrophil recruitment. This model has previously been published by us and others. The mechanisms of bacterial dissemination and/or bacterial pathogenicity was not the aim of this study. Focusing on the molecular mechanisms of chemokine transcytosis and the regulation of neutrophil recruitment, this study and the results of previous reports (e.g. Rossaint, J Exp Med, 2021;218(7):e20201353; Lee, FEMS Microbiol Lett 2012;331(1):17-24) clearly show the induction of a robust neutrophil activation and recruitment in response to pulmonary infection with the ATCC 13883 Kpn strain, thus rendering it suitable for use in our model with the given scientific question. In addition, we now also include data from survival studies as well as data using a different pathogen (*E. coli*) clearly show that the detected mechanisms are a general pathobiological process underlining the relevance of our findings (see Figure 2).

We revised Figures 1-2 and now present the CFUs in standard log scale mode. Furthermore, we performed additional experiments with blocking antibodies against CXCR2 *in vivo* and added the bacterial CFU counts for the experiments in Figure 2J-K.

The following paragraph was added to the results section (page 6, line 1):

“Intratracheal instillation of *K. pneumoniae* resulted in recruitment of neutrophils into the lungs of WT control mice compared to saline-treated control mice and was significantly reduced in all CXCR2 knockout strains tested and WT that received a blocking CXCR2 antibody intratracheally (Figure 1A, D, G and Supplemental Figure 5A-H). The baseline and post-infection experiments were conducted concurrently. Impaired neutrophil recruitment in knockout mice and after intratracheal instillation of a blocking CXCR2 antibody was observed in the whole lung tissue and also in the bronchoalveolar lavage (BAL, Figure 2A, D, G). In all knockout mouse strains and WT mice who received a blocking CXCR2 antibody intratracheally, a decrease in neutrophil recruitment was associated with a significant increase in bacterial burden, as e.g. determined by the quantification of bacterial colony forming unites (CFUs), in the lung and the spleen (Figure 1B, C, E, F, H, I and Supplemental Figure 5A-H). This effect was further evident in the BAL and the blood (Figure 2B, C, E, F, H, I).”

8) Fig. 3: From the provided photographic illustrations, it appears that alveolar septa are strongly enlarged in both WT and CXCR2 KO mice challenged with either CXCL1 or fMLP. How does this affect the transit time of PMN crossing the barrier to reach the alveoli, and thus being accessible by BAL? Inclusion of control sections of untreated mice is needed.

Response: We thank the reviewer for this valuable comment and measured the thickness of alveolar septa in all groups. In addition, we included control sections of untreated control mice.

The following paragraph was added to the results section (page 7, line 26):

“Both intratracheal CXCL1 and fMLP instillation did not significantly alter the thickness of the alveolar septa in the lung compared to saline-treated control mice after 4 hours (Supplemental Figure 9A).”

9) Did the authors determine CXCL1 levels in plasma of infected WT and global versus endo-/epithelial CXCR2 deficient mice? A global (or partial) chemokine receptor KO should lead to increased levels of the ligand in the circulation upon infection. Against this, why should intratracheal CXCL1 be transcytosed across barrier cells into the circulation despite the fact that just WT mice express CXCR2 on both barrier cell types, which should capture and sub-sequently internalize the ligand?

Response: We agree with the reviewer and performed additional experiments to analyze the levels of CXCL1 in the serum, lung tissue and BAL of global CXCR2^{-/-} mice and WT mice under baseline conditions and following the induction of bacterial pneumonia.

The following paragraph was added to the results section (page 7, line 7):

“Furthermore, we analyzed CXCL1 chemokine levels in plasma, BALF and lung homogenates obtained from *K. pneumoniae*-treated mice and found significantly increased chemokine levels upon pneumonia compared to baseline conditions (Supplemental Figure 8G-I). Interestingly, CXCL1 levels were increased in plasma and BALF and decreases in lung homogenisates from CXCR2^{-/-} mice compared to WT mice (Supplemental Figure 8G-I).”

The following paragraph was added to the discussion (page 16, line 13):

“We also demonstrate that administration of a relatively small exogenous amount of the chemokine CXCL1 is sufficient to trigger the immediate recruitment of circulating neutrophils into the non-infected “resting” lung parenchyma. Interestingly, CXCL1 levels in the circulation were found to be increased in CXCR2-deficient mice, resembling an overshooting immune system activation in the circulation, which is also indicated by the increased CFU counts in the blood of these mice. This indicates that early neutrophil recruitment into the lung, mediated by a relatively small amount of transcytosed CXCL1 originating from the alveoli, is necessary to efficiently contain bacterial infections and subsequent pathogen dissemination into the circulation.”

10) Given the published cross-reactivity between CC and CXC chemokine receptors in inflammatory monocyte and neutrophil migration to the lung, what effect did the authors note regarding inflammatory monocyte trafficking in the current model?

Response: We thank the reviewer for highlighting this interesting aspect and performed additional experiments.

The following paragraph was added to the results section (page 7, line 2):

“To investigate a possible effect of CXCR2 knockout on CC chemokine homeostasis, we investigated trafficking of inflammatory monocytes in the lung in saline-treated control animals and after induction of pneumonia and did not observe significant differences (Supplemental Figure 8A-F).”

11) The authors suggest a severe impairment of lung bacterial clearance in selective CXCR2 KO, but this has not been proven in respective survival studies or more detailed infection studies addressing the course of infection over time. This makes the biological significance of the reported findings enigmatic.

Response: We agree with the reviewer and performed a survival analysis.

The following paragraph was added to the results section (page 6, line 18):

“Administration of a higher CFU of *K. pneumoniae* in the intratracheal inoculum led to a significantly decreased survival of CXCR2^{-/-} compared to WT control mice (Figure 2J).”

Reviewer 1

In the second paragraph before introducing ACKR (Atypical Chemokine Receptor 1), the authors should provide a brief description of current knowledge on chemokine transcytosis. This will help to make the narrative more coherent and easier for general readers to understand the study's background. Additionally, they should spell out ACKR1 on its first mention.

Response: We thank the reviewer for the valuable suggestion and added the following paragraph to the introduction (page 3, line 22):

“Chemokine transcytosis during acute inflammation involves the transport of chemokines across endothelial barriers to facilitate immune cell recruitment in response to inflammatory stimuli. Chemokine binding to classical 7-transmembrane receptors (7TMR) may trigger G-protein-coupled activation of intracellular signaling cascades. In contrast, non-signaling 7TMRs do not activate G-protein-coupled signaling but may foster chemokine binding and receptor-mediated endocytosis, involving activation of MAP kinases and β -arrestins (Rajagopal, PNAS 2010). Once internalized, chemokines undergo intracellular trafficking and vesicular transport across the endothelial barrier, a process known as transcytosis. Upon reaching the tissue space, they establish gradients that guide leukocytes to the site of inflammation, amplifying the immune response.”

Reviewer 2

The authors have performed extensive additional experiments to i) demonstrate that CXCL1 transport occurs via the trans- rather than the paracellular route, ii) on the effects of different inhibitors of transcytotic transport in the transwell bilayer model, iii) in bone marrow chimeric Btk-KO mice, and iv) in a second model of E. coli induced bacterial pneumonia. These additional data largely confirm the authors' proposed concept, and have significantly strengthened the paper. A few major and minor concerns, however, persist or arose de novo as a result of the revised graphs and text, and need to be addressed.

Major comments:

1. In response to my previous comment 3 "Are expression levels of CXCR2 in lung endothelial or epithelial cells differentially regulated in pneumonia?" the authors now state that CXCR2 expression was found to be increased in WT mice after pneumonia induction and refer to Supplemental Fig. 1; however, I was unable to detect any flow cytometric data from pneumonia mice in the respective figure.

Response: We thank the reviewer for pointing this out and now included the respective data in Supplemental Figure 1.

2. In response to my previous comment 4, the authors replaced the original exemplary flow cytometric data, but unfortunately these new data leave me even more confused: From what I see in Suppl. Fig. 1, it seems that CXCR2 levels decreased in BOTH endothelial and epithelial cells no matter whether the knockout was seemingly specific for endothelial or epithelial cells (notably, the strongest loss of CXCR2 is seen in endothelial cells after knockout of epithelial CXCR2 (panel on the lower left)). This critical aspects needs proper clarification.

Response: We thank the reviewer for the careful appreciation of our data and apologize for the misleading data presentation, which was due to an error in the arrangement of the figure panel. We have revised Supplemental Figure 1 and now show the correct histograms.

3. I am also still confused about the intravital microscopic assessment of "transmigrating neutrophils" (page 7, line 23). As stated now by the authors "intravital microscopy does not allow the precise allocation of the neutrophils within the different compartments of the lung. Therefore, we analyzed the presence of interacting neutrophils in the lung as cells remaining stationary for >30s in the pulmonary vasculature, the interstitial space or the alveoli." First, how can you differentiate between stationary cells in the pulmonary vasculature, the interstitial space or the alveoli, when you cannot precisely allocate neutrophils to these different compartments? Second, how does the number of stationary neutrophils translate into counts of "transmigrating neutrophils"? Finally, the authors report that "neutrophils were visualized in the alveoli by intravenous administration of an AF488-coupled Gr-1 antibody before starting microscopy" – how can the antibody reach the neutrophil if the neutrophil is already in the interstitium or the alveolar space?

Response: We thank the reviewer for pointing out this important aspect. We performed additional experiments by using an *ex vivo* model of viable precision cut lung slices obtained from our different CXCR2-deficient mouse strains combined with spinning disc confocal

microscopy and 3D image reconstruction. The following paragraph has been added to the results section (page 9, line 19):

“To visualize the precise allocation of neutrophils in different compartments of the lung, we performed spinning disc confocal microscopy of viable precision cut lung slices *ex vivo* (Supplemental Figure 9G-H). These data demonstrate that CXCR2-deficiency in lung epithelial or endothelial cells leads to a decreased emigration of neutrophils from the vascular compartment into the lung tissue (Supplemental Figure 9I).”

Minor comments

1. *In the new fig. 6G, please indicate in the graph which of the x-axis denominations (+/+, -/-) refer to endothelial and which to epithelial cells.*

Response: We apologize for the mistake and added the missing x-axis denominations to Figure 6G.

2. *The authors have added new data to explore the regulation of CXCL1 transcytosis by Btk in greater depth and now show regulation of p38 and Akt phosphorylation by Btk. To make the mechanistic connect, could the authors add a sentence or two how these pathways link to clathrin-coated pits mediated transcytosis?*

Response: We thank the reviewer for this suggestion and added the following paragraph to the discussion (page 18, line 19):

“Both p38 MAPK and Akt signaling pathways may regulate clathrin-coated pit (CCP)-mediated transcytosis by modulating vesicle formation, cargo recruitment, and cytoskeletal dynamics. p38 MAPK enhances dynamin activity, facilitating vesicle budding, and regulates actin cytoskeleton remodeling via phosphorylation of heat shock protein 27 (HSP27), which supports actin-dependent vesicle movement (Guay, J Cell Sci 1997; Rousseau, Oncogene 1997). Additionally, p38 influences Rab GTPases involved in vesicle sorting and maturation after endocytosis. Akt signaling contributes by regulating adaptor proteins such as AP-2, which are essential for cargo selection and clathrin recruitment, and by modulating Rab proteins like Rab11 to direct vesicles through the transcytotic pathway (Schenck, Cell 2008). Together, p38 and Akt may coordinate the internalization, trafficking, and exocytosis of CXCL1, with p38 supporting vesicle scission and actin remodeling, while Akt enhances cargo selection, sorting, and vesicle fusion, ensuring efficient CXCL1 transcytosis.”

3. *Graphical abstract: In the graphical abstract, it seems as if CXCR2 once it binds CXCL1 is transported across both the epithelium and endothelium. This is likely not the case; rather, CXCR2 will release CXCL1 on the basolateral side of the epithelium where it is then taken up in turn by CXCR2 expressed on the apical side of the endothelium and transcytosed again to be expressed on the vascular surface.*

Response: We thank the reviewer for pointing out this aspect and revised the graphical abstract accordingly.

4. Page 14, line 11: Please change “contend” to “content”

Response: We thank the reviewer for the careful reading of our manuscript and corrected the mistake accordingly.

Reviewer 3

The manuscript, “Alveolar epithelial and vascular CXCR2 mediates transcytosis of CXCL1 in inflamed lungs” uses comparisons between global CXCR2 vs epithelial CXCR2 (Shh Cre) vs endothelial CXCR2 (Cdh5 Cre) to indicate a novel lung-specific form of chemokine transcytosis dependent on CXCR2 expression in non-hematopoietic cells. They primarily use a model of Klebsiella infection in alveolae to investigate CXCL1 distribution and neutrophil recruitment in acute lung inflammation. This work is interesting and builds upon a small but growing body of evidence that chemokine receptors have non-signaling roles involved in chemoattractant distribution. After revisions addressing concerns, publication is recommended after addressing 1 major issue and several minor issues:

1. In Figure 6G, inhibitors for clathrin-, caveolin-, and dynamin-dependent endocytosis/transcytosis are used with the intention to further the transcytosis mechanism used by CXCR2 in the transport of CXCL1. However, the data used for this experiment measures neutrophil recruitment, not CXCL1 distribution. Thus, the data does not directly show that clathrin is necessary for CXCL1 transcytosis rather than neutrophil infiltration. Please directly show CXCL1 changes in distribution using the inhibitors.

Response: We agree with the reviewer and analyzed the CXCL1 levels in the supernatant to show CXCL1 translocation. The following paragraph was added to the results section (page 12, line 5):

“To further elaborate on the exact organellar route of transcytosis, we performed transmigration assays using the transwell bilayer model with inhibitors of clathrin- (pitstop 2), caveolin- (MBCD) or dynamin-mediated transcytosis (dyngo 4a) and observed a significant reduction in neutrophil transmigration after inhibition of clathrin-mediated transcytosis (Figure 6G) as well as reduced CXCL1 levels in the upper well above the endothelial cell layer (Figure 6H).”

2. In Figures 1, 2, and Supplementary Figure 4, please delineate which bars are saline versus infection on the graphs themselves.

Response: We thank the reviewer for this suggestion and now delineate which bars are saline or infection with Klebsiella in the graphs themselves of Figures 1 and 2. However, Supplementary Figure 4 refers to the *in vitro* expression analysis of surface markers of endothelial and epithelial cells isolated from mice, and thus no saline vs. infection treatment was performed in these groups.

3. Address in the discussion an evolution theory/hypothesis as to why CXCR2 on lung endothelial cells and not CXCR2 in the kidney or cremaster are privileged to transcytose CXCL1.

Response: We thank the reviewer for highlighting this aspect and included the following paragraph in the discussion (page 16, line 16):

“Why CXCR2 in the lung would be privileged to transcytose CXCR2 over other receptors, e.g. ACKR1, remains elusive. One aspect may be the constant exposure of the large internal lung surface to air-borne pathogens requiring a very efficient first line of innate immune defense, but more detailed investigations may be required to address this specific question.”

Reviewer 4

Response: We thank the reviewer for evaluation of our work.

Reviewer 5

My primary concerns have not been addressed:

1) The concern of using the anti-Gr-1 antibody clone RB6-8C5 for depletion of neutrophils was related to the previously reported co-depleting activities of this antibody against monocytes. Therefore, it simply makes no sense to perform immunofluorescence analysis of lung tissue collected from mice pre-treated with this antibody while stating that no monocytes were found in these mice at 4 h post-infection. Rather, FACS analyses of lung tissue digests would have been needed to clarify the effect of this antibody on lung monocyte subsets in the given experimental context.

Response: We thank the reviewer for the clarification of this points and performed additional experiments to address the reviewer's concerns. The following paragraph has been added to the results section (page 9, line 12):

“To exclude that neutrophil depletion using the anti-Gr1 antibody clone RB6-8C5 causes considerable co-depletion of monocytes, we injected CXCL1 intratracheally, administered the Gr1 antibody or isotype control and analyzed the abundance of neutrophils and monocytes in the lung after 4 hours (Supplemental Figure 9C-D). Furthermore, the administration of labeled Gr1 antibody did not significantly decrease neutrophil and monocyte blood counts in WT mice after the early observation period of 4 hours (Supplemental Figure 9E-F).”

2) The biological relevance of the employed model is still not clear. The authors provide survival curves comparing WT with CXCR2 KO mice after challenge with K. pneumoniae. However, this experiment was not asked for, as it simply confirms findings that are already known, i.e. that CXCR2 is relevant for survival of bacterial pneumonia. However, the question was whether selective endo- and epithelial CXCR2 KO would also be relevant to survival of pneumonia? Just these critical survival studies have not been provided by the authors.

Response: We agree with the reviewer and performed additional experiments analyzing survival also in conditional CXCR2^{fl/fl}Cdh^{Cre+} and CXCR2^{fl/fl}Shh^{Cre+} mice. The following paragraph was added to the results section (page 7, line 18):

“Administration of a higher CFU of K. pneumonia in the intratracheal inoculum led to a significantly decreased survival of CXCR2^{-/-} as well as conditional CXCR2^{fl/fl}Cdh^{Cre+} and CXCR2^{fl/fl}Shh^{Cre+} mice specifically lacking CXCR2 in endothelial or epithelial cells compared to the respective control mice (Figure 2J).”

3) I was asking for CXCL1 chemokine levels in plasma of CXCR2 KO and partial CXCR2 KO mice (i.e., endo- versus epithelial CXCR2 KO). However, the authors just report plasmatic CXCL1 levels in global CXCR2 KO mice challenged with K. pneumoniae. Of note, these data do not add anything novel to the study, nor support the claims of the authors. It is well known that global knockout of a given chemokine receptor increases plasmatic levels of the respective chemokine ligands in infected mice. Considering endothelial CXCR2 KO mice, the concept would be that endothelial CXCR2 deficiency would lead to increased plasmatic CXCL1 levels, which may cause CXCR2 receptor desensitization on the neutrophil surface (which can be determined by FACS analysis) followed by receptor internalization, thereby rendering neutrophils unresponsive to local alveolar CXCL1 chemokine levels. This in turn would

promote bacterial pneumonia and ultimately bacterial dissemination and mortality. However, these kind of experiments have not been addressed during revision of the manuscript.

Response: We thank the reviewer for pointing out these important aspects. We have performed additional experiments. The following paragraph has been added to the results section (page 8, line 12):

“Interestingly, CXCL1 levels were increased in plasma and BALF from CXCR2^{-/-}, CXCR2^{fl/fl}Cdh5^{Cre+} and CXCR2^{fl/fl}Shh^{Cre+} mice compared to control mice (Supplemental Figure 8G-L). Endothelial CXCR2 deficiency led to CXCR2 desensitization by decreased CXCR2 expression on neutrophils (Supplemental Figure 8M) and subsequently impaired neutrophil migration 24 hours after the induction of pneumonia (Supplemental Figure 8N-P).”